# Boundary layers, transport and universal distribution in boundary driven active systems

**Pritha Dolai[1,2,3†] and Arghya Das[4⋆]**

**1** National Institute of Technology Karnataka Surathkal, Mangalore 575025, India
**2** Friedrich-Alexander-Universität Erlangen-Nürnberg, Erlangen 91054, Germany
**3** Max-Planck-Zentrum für Physik und Medizin, Erlangen 91058, Germany
**4** The Institute of Mathematical Sciences, Taramani, Chennai, 600113, India

⋆ arghyaburo21@gmail.com , † pritha@nitk.edu.in

## Abstract

We discuss analytical results for a run-and-tumble particle (RTP) in one dimension in presence of boundary reservoirs. It exhibits 'kinetic boundary layers', nonmonotonous distribution, violation of Fourier's law, diffusion facilitated current reversal and optimisation on tuning dynamical parameters, and a Seebeck-like effect in the steady state. The spatial and internal degrees of freedom together possess a symmetry, using which we find the eigenspectrum for large systems. The eigenvalues are arranged in two bands which can mix in certain conditions resulting in a crossover in the relaxation. The late time distribution for large systems is obtained analytically; it retains a strong and often dominant 'active' contribution in the bulk rendering an effective passive-like description inadequate. A nontrivial 'Milne length' also emerges in the dynamics. Finally, a novel universality is proposed in the absorbing boundary problem for dynamics with short-ranged colored noise. Active particles driven by active reservoirs may thus provide a common physical ground for diverse and new nonequilibrium phenomena.

# 1 Introduction

Systems with self-propelled or active particles became a paradigm of nonequilibrium processes. Active agents consume energy from environment and convert it into mechanical work to propel themselves. Constant energy consumption drives active systems far from equilibrium [1, 2, 3]. There are numerous real life examples of active systems across scales including bacterial colony, school of fish, flock of birds, janus particles[4, 5], vibrated granular rods and disks [6, 7] and many more. Such systems exhibit a plethora of nontrivial collective phenomena, e.g. pattern formation, motility-induced phase separation and clustering [2, 3, 8, 9, 10], giant number fluctuations [11], casimir effect [12]. They reveal rich nonequilibrium properties even at the single particle level: examples include non-Boltzmann steady-state, accumulation near boundary, shape transition in probability distribution in a trapping potential, a condensation-like transition in the free motion with random activity [13, 14, 15, 16, 17, 18]. Active motion is modelled with a Langevin-like equation with an 'active noise', which is typically coloured and violates the fluctuation-response relation, leading to such novel behaviour. In literature there are three widely studied scalar active particle models: run-and-tumble particles (RTP), active Brownian particles (ABP) and active-Ornstein-Uhlenbeck particles (AOUP).

Active particles show interesting behaviour in the presence of drives, thermal noise, obstacles and different boundary conditions. For example, RTPs subject to spatially periodic drive go through a transition from non-ergodic trapped states to moving states [19]. In presence of shear forces in two dimension, underdamped ABPs develop a boundary layer and flow reversal occurs near the wall [20]. Confining boundaries are known to have strong effects on active systems, e.g. boundary accumulation and boundary layer formation [13, 21], long-range effects and system size dependent behaviour [22, 23, 24, 25, 26, 27], etc. Interestingly, the nature of the boundary layer and the pressure exerted by the particles on a wall depends on the shape of the wall [21] as well as on the interactions [22]. Thermal fluctuations, ubiquitous but usually neglected, can have nontrivial effects on active motion e.g. formation of bound states [28], current modulation on a ratchet [29], and triggering phase separation [30, 31].

In this work we focus on boundary-driven processes. Systems connected to particle or energy reservoirs have been studied for centuries, but there is a renewed interest in these processes particularly in the context of low dimensional transport. The canonical picture is given by Brownian motion in a finite system connected to particle baths: the steady state density profile

is a linear interpolation of the densities at the boundaries [32] while the current is proportional the density gradient as given by the Fick's law. However in the presence of interaction, bulk drive, or more than one conserved quantities etc. richer and often nonintuitive behaviour is observed. Models of boundary driven interacting diffusive systems exhibit long ranged correlation [33, 34, 35]; introduction of a bulk drive gives rise to different phases [36, 37]. The thermal transport properties become anomalous and often violate Fick's law in one dimensional interacting anharmonic chains [38, 39, 40]. Nonlinear bulk density profile and nongradient steady state current emerges in simple noninteracting cases where the particle dynamics is governed by strongly correlated noise, for instance Levy walk connected to particle reservoirs [41, 40]. Violation of the Fourier's law i.e. a finite thermal conduction with zero temperature gradient was demonstrated for nonlinearly coupled passive systems driven by non-Gaussian white noises [43, 42]. New effects induced by 'active heat baths' in a passive harmonic (linearly coupled) chain, like current reversal in the energy transport and boundary features in the steady state, have also been reported recently [44, 45]. In contrast, results for active systems in presence of boundary reservoirs are rare[46], and to our knowledge the effects of active reservoirs are not yet discussed. It is natural to ask what are the steady state, transport properties, dynamical fluctuations and relaxation behaviour of different boundary driven active systems and how do they differ across models and from their passive counterpart. To explore the nonequilibrium physics in this class of processes, we consider an elementary model amenable to simple analysis and exact results.

In this paper we study non-interacting free RTPs in presence of thermal noise in a one-dimensional channel connected to particle reservoirs at both ends maintaining fixed particle densities and magnetisation [1]. The results are surprisingly rich and intriguing. The solution for steady state and large time distribution show the existence of 'kinetic boundary layers' [47] characterised by exponentially decaying components in the density profile near the reservoirs. These are distinct from the usual boundary layers reported earlier for confining walls or obstacles in that, here these occur in the presence of particle fluxes accross the boundaries and can correspond to both accumulation and depletion. For zero boundary magnetisation, the steady state current and bulk density profile show a passive-like behaviour while the effect of activity which is notable for large persistence, becomes prominent at the boundary layer. However, the signature of activity appears in the bulk through a finite magnetisation. Remarkably, in the presence of nonzero boundary magnetisations, the density profile can become nonmonotonic, there is current without density gradient, and in particular a diffusion-facilitated current reversal and current maximisation upon tuning the particle parameters can take place. In this connection a Seebeck-like effect is identified, with boundary magnetisations taking the role of temperatures and density differences being analogous to voltage differences.

The time-dependent solution to the boundary value problem is identical to that of the absorbing boundary problem which is relatively less discussed [48, 49, 50]. We note that the boundary condition endows the system with a 'reflection symmetry'. Using this, the eigenspectrum and time-dependent solutions are obtained analytically for large system-sizes. The eigenvalues of the time evolution operator is arranged in two distinct bands separated by the tumble rate. The relaxation is diffusive for large $L$; as the system size is reduced the bands start overlapping, and for small enough systems the relaxation rate crosses over to an $L$-independent value. Such crossovers had been found earlier for one and two particle systems on a ring[51, 28]. We observe that in each of the eigenstate the 'active' contribution appears only as $O(L^{-1})$ correction to the leading 'passive-like' term for large $L$; but, contrary to expectations, at large times both the active and passive contributions occur at the same order in the bulk. For higher persistence,

---

[1]Since the RTPs carry two internal states in 1-dimension, the magnetisation is the difference of densities of the two species.

the large distance late time distribution and related properties are in fact dominated by the active contribution. Interestingly, although the distribution satisfies the absorbing condition at the boundaries in the presence of thermal noise, a 'Milne length' [52, 53, 54] is still identified in the distribution. This length has also been found in the context of survival probability of random flights [55] and athermal RTPs [49]. In the present case it dependends nontrivially on the diffusivity and other dynamical parameters. We found rather non-intuitively that the Milne length has a significant role in the the steady state. A kinetic boundary layer related to the Milne length appears in the time-dependent solution as well and this is argued to be a consequence of thermal diffusion. We further argue that this is related to a net imbalance of the two species of particles in the system, which goes away as soon as the diffusivity vanishes. As a whole, the interplay of thermal noise and active motion causes important ramifications in the dynamics in the presence of boundary driving.

It turns out that many of the said features obtained for RTPs also appear in the cases involving underdamped passive particle, athermal AOUP, and ABP, and are possibly generic to dynamics with coloured noise. In the absorbing boundary problem the emergence of kinetic boundary layers and Milne length were known long back for the underdamped passive motion [54, 56, 57, 58]. In that case the large time density profile is in fact quantitatively similar to that obtained here for RTPs. It prompts us to propose that the late time distribution away from the absorbing wall is independent of the realisation of the coloured noise, provided that the noise correlation is short ranged.

Rest of the paper is organized as follows. In Section 2, we describe the model system of a boundary-driven run-and-tumble particle. The steady-state and the time-dependent solutions are presented in Sections 3 and 4, respectively. A universality in the large time distribution is proposed in Section 5. In Section 6 we summarise our perspective and conclude.

## 2 Model: boundary driven RTPs

We consider a run-and-tumble particle (RTP) moving in a finite one-dimensional box of length $L$ subject to a translational noise $\eta(t)$. The box is connected to particle reservoirs at the two ends. We assume that the microscopic details of the interaction of the reservoir with the system at the boundary is not important except that it provides a fixed boundary condition. The equation of motion of the particle is,

$$\dot{x} = v\sigma(t) + \eta(t), \tag{1}$$

where $v$ is the self-propulsion speed, $\sigma(t)$ is the intrinsic orientation (spin) characterized by a telegraphic noise that takes values $\pm 1$ and it flips between the two states with rate $\omega$, and $\eta(t)$ is the Gaussian white noise with $\langle \eta(t) \rangle = 0$, $\langle \eta(t)\eta(t') \rangle = 2D\delta(t - t')$, $D$ being the diffusion constant.

The run and tumble motion described in terms of $(x(t), \sigma(t))$ is a Markov process. The state of the particle at a time $t$ is given by, $|P(x,t)\rangle = \begin{bmatrix} P_+(x,t) \\ P_-(x,t) \end{bmatrix}$, where $P_+(x,t)$ and $P_-(x,t)$ are the probabilities to find a particle at x at time t with instantaneous spin $+1$ and $-1$ respectively. $P_+$ and $P_-$ satisfy the Master equations,

$$\partial_t P_+ = D\partial_x^2 P_+ - v\partial_x P_+ - \omega P_+ + \omega P_-, \tag{2a}$$

$$\partial_t P_- = D\partial_x^2 P_- + v\partial_x P_- + \omega P_+ - \omega P_-, \tag{2b}$$

with the boundary conditions,

$$\text{At } x = 0: \ P_+(0,t) = P_0^+, \ P_-(0,t) = P_0^- \ ;$$
$$\text{At } x = L: \ P_+(L,t) = P_1^+, \ P_-(L,t) = P_1^- \ . \tag{3}$$

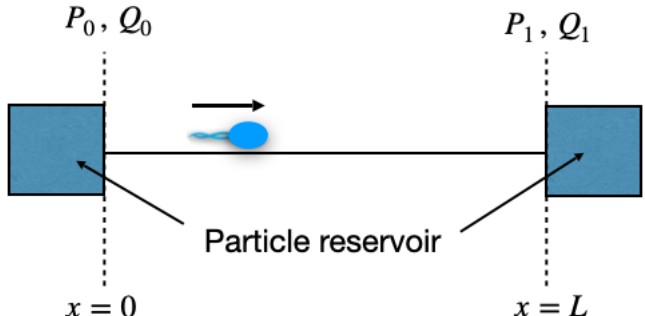

Figure 1: Schematic of the boundary driven active system. Each RTP moves on a one-dimensional line bounded between $x = 0$ and $L$. The system is connected to particle reservoirs at both the ends maintaining fixed particle densities $P_0$ and $P_1$ and the corresponding particle magnetisation $Q_0$ and $Q_1$ at $x = 0$, $L$ respectively.

$P_0^{\pm}$ and $P_1^{\pm}$ are the probability densities of each species of particles at the two boundaries $x = 0$ and $x = L$ respectively. The total probability density is $P(x,t) = P_+(x,t) + P_-(x,t)$. We also define a quantity $Q(x,t) := \langle \sigma \rangle_{(x,t)} = P_+(x,t) - P_-(x,t)$ which is analogous to magnetization density. In terms of $P$ and $Q$ the boundary conditions are rewritten as,

$$\begin{aligned} \text{At } x = 0: \ & P(0) = P_0, \ Q(0) = Q_0, \\ \text{At } x = L: \ & P(L) = P_1, \ Q(L) = Q_1. \end{aligned} \tag{4}$$

The schematic of the setup is shown in Figure 1.

## 3    Steady state distribution

In the steady state $\partial_t P_+ = \partial_t P_- = 0$. Using the definitions of $P$ and $Q$, the equations for the steady state can be written as,

$$\partial_t P = D\partial_x^2 P - v\partial_x Q = 0 \tag{5}$$

$$\partial_t Q = D\partial_x^2 Q - v\partial_x P - 2\omega Q = 0. \tag{6}$$

For arbitrary densities of the two species of particles at each of the boundaries, we need to solve Eqs. (5)-(6) with the boundary conditions given in Eq. (4). The solution for the density and magnetisation profiles and the current $J_0 = -D\partial_x P + v\,Q$ in the steady state are,

$$\begin{aligned} P(x) = P_0 + M &\left[ B_0 - \Delta Q\, \frac{e^{-\mu L}\left(1 + e^{-\mu L}\right)}{(1 - e^{-\mu L})^2} \right] + \frac{\Delta P - M(Q_0 + Q_1)}{M\frac{v}{\omega} + L}\, x \\ &- \frac{v}{\mu D}\, \frac{B_0\left(e^{-\mu x} - e^{-\mu(L-x)}\right) - \Delta Q\, e^{-\mu(L-x)}}{1 + e^{-\mu L}}\,, \end{aligned} \tag{7}$$

$$Q(x) = -\frac{\Delta P - M(Q_0 + Q_1)}{2(M + L\frac{\omega}{v})} + \frac{B_0\left(e^{-\mu x} + e^{-\mu(L-x)}\right) + \Delta Q\, e^{-\mu(L-x)}}{1 + e^{-\mu L}}\,, \tag{8}$$

$$J_0 = -\frac{D_e}{M\frac{v}{\omega} + L}\left(\Delta P - M(Q_0 + Q_1)\right), \tag{9}$$

where, $\Delta P = P_1 - P_0$, $\Delta Q = Q_1 - Q_0$, $D_e = D + \frac{v^2}{2\omega}$, $\mu = \frac{\sqrt{2\omega D_e}}{D}$, $M = \frac{v}{\mu D}\frac{1 - e^{-\mu L}}{1 + e^{-\mu L}}$, and $B_0 = \frac{\Delta P - M(Q_0 + Q_1)}{2(M + L\frac{\omega}{v})} + Q_0 - \Delta Q\, \frac{e^{-\mu L}}{1 - e^{-\mu L}}$. In the $D = 0$ case $\mu$ diverges and the exponential terms vanish. This gives rise to discontinuities in the density and magnetisation profiles at the boundaries and it is smoothened to kinetic boundary layers for nanvanishing diffusion. We list a few implications of the Eqs. (7)-(9).

I. For $L$ large the current and the density gradient in the bulk ($\mu^{-1} \ll x \ll L$) are proportional to $(L + 2\,l_M)^{-1}$, where $l_M = \frac{v^2}{2\omega\,\sqrt{2\omega D_e}}$ is the Milne length as shown later. This increase in the effective system length suggests that the system in the bulk pertains to boundary conditions at a distance $l_M$ beyond the actual boundaries, as in several absorbing boundary problems. This picture however holds only for $Q_0 = Q_1 = 0$.

II. The bulk density profile, when extended to the boundaries, violates the boundary condition by an amount proportional to $M$ (or $l_M$). For $D > 0$ 'fine-tuned' kinetic boundary layers representing particle accumulation at one end and depletion at the other are formed which restore the boundary conditions. Further, the density profile may become nonmonotonic near the boundaries.

III. The current is finite even if $\Delta P = 0$: $J_0 \propto (Q_0 + Q_1)/(L + 2\,l_M)$, in violation of Fourier's law. This suggests that the boundary magnetisations effectively act as drives inducing a constant current in the system in the absence of a density gradient. However this similarity cannot be exact, since in the present case the current gives rise to a linear density profile away from the boundaries, unlike the usual driven passive case in which density remains uniform.

IV. On the contrary, with suitable boundary magnetisations such that $Q_0 + Q_1 = \Delta P/M$, the current can be made to vanish without invoking any external bias to counteract the density gradient. In large systems this leads to a uniform density profile and almost vanishing magnetisation in the bulk and the inhomogenities remain only in the boundary layers. Imposing an additional condition of zero density gradient ($\Delta P = 0$) mimics the steady state result for reflecting boundaries [13].

V. In the athermal ($D = 0$) case, $M = 1$ irrespective of the dynamical parameters. Thus $J_0 \propto (\Delta P - \overline{Q_0 + Q_1}) = P_1^- - P_0^+$, implying that the direction of the current is determined entirely by the boundary conditions. The presence of thermal noise drastically alters the picture. For any $D > 0$, either of the activity parameters $v, \omega$ can be tuned to access all values of $M \in (0, 1)$. Consequently, for a fixed boundary condition with the only constraint $P_1^- < P_0^+$ i.e. $0 < \frac{\Delta P}{Q_0 + Q_1} < 1$, we can always tune the parameters across $M^* = \frac{\Delta P}{Q_0 + Q_1}$ and a current reversal takes place.

VI. The current reversal, originating either from change in the boundary conditions or tuning of the dynamical parameters, leads to a sign flip in the slope of the bulk density profile. The bulk magnetisation being proportional to the current also changes sign. This suggests that the current governs the nature of the bulk profiles and not the boundary conditions.

VII. In the zero current condition, e.g. in a 1D box from where particles cannot escape, maintaining a nonzero net boundary magnetisation induces a global density difference, $\Delta P = M(Q_0 + Q_1)$. This phenomenon is reminiscent of the Seebeck effect with coefficient $M$ and the boundry magnetisations mimicking heat baths.

VIII. For $D > 0$ interesting nonmonotonicities are present in the current. A *maximal* current occurs in two distinct conditions in the large $L$ limit. Defining $D_a = \frac{v^2}{2\omega}$, the expression of the current in the thermodynamic limit becomes, $J_0 \approx -\frac{D + D_a}{L}[\Delta P - \sqrt{\frac{D_a}{D + D_a}}\,(Q_0 + Q_1)]$ with $M \approx \sqrt{\frac{D_a}{D + D_a}}$. We can see that the competing currents due to the density gradient and boundary magnetisations are both enhanced but in a different manner as $D$ and $D_a$ are increased. The extremal current for fixed activity parameters and changing thermal noise occurs at $\frac{\partial J_0}{\partial D} = 0$, which translates to $M = M_p \equiv 2\frac{\Delta P}{Q_0 + Q_1}$ under the condition $0 < \frac{\Delta P}{Q_0 + Q_1} < \frac{1}{2}$. This condition also allows for the current reversal, implying that an optimal current can be achieved and the direction can be controlled by tuning $D$ only.

On the other hand, the extremal current condition for changing activity parameters is given by $\frac{\partial J_0}{\partial D_a} = 0$, which implies, $M = M_a \equiv \frac{\Delta P}{Q_0+Q_1} - \sqrt{(\frac{\Delta P}{Q_0+Q_1})^2 - 1}$. This occurs only when $\frac{\Delta P}{Q_0+Q_1} > 1$, which does not allow for a current reversal. Thus the conditions for optimal current clearly distinguishes the role of thermal diffusivity and the effective diffusivity due to activity. For $\frac{1}{2} < \frac{\Delta P}{Q_0+Q_1} < 1$ the current is monotonous for all parameter values and only a current reversal occurs at $M^*$.

The distributions and the current reversal are shown in Figure 2. The nonmonotonicities in the current are shown in Figure 3 [2].

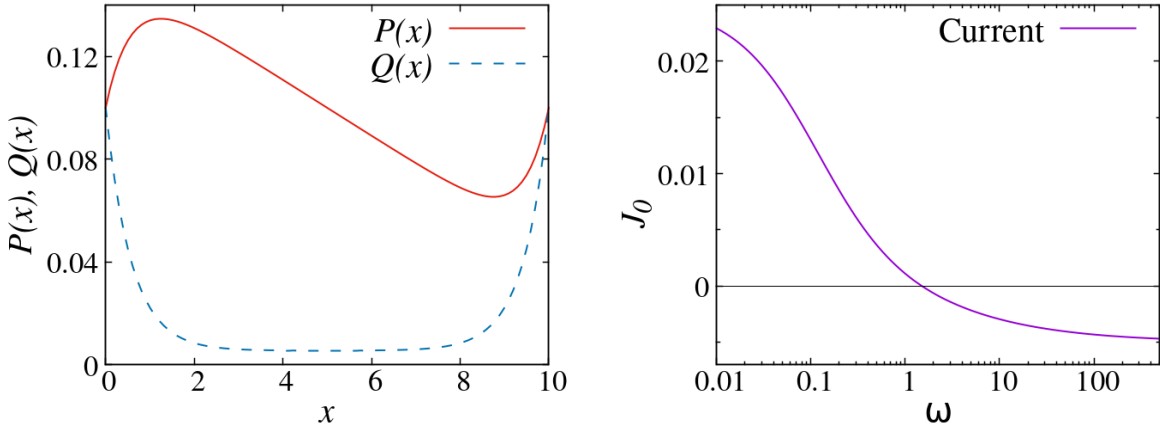

Figure 2: Left panel: Density (solid line) and magnetisation (dashed line) profiles in the steady state of the boundary driven RTP. Here $v = 1.0, D = 1.0, w = 1.0, L = 10.0$ and $P_0 = P_1 = 0.1, Q_0 = Q_1 = 0.1$. Right panel: Current reversal as the tumble rate $\omega$ is tuned. Here $v = 1.0, D = 1.0, L = 10.0$ and $P_0 = 0.075, P_1 = 0.125, Q_0 = Q_1 = 0.05$.

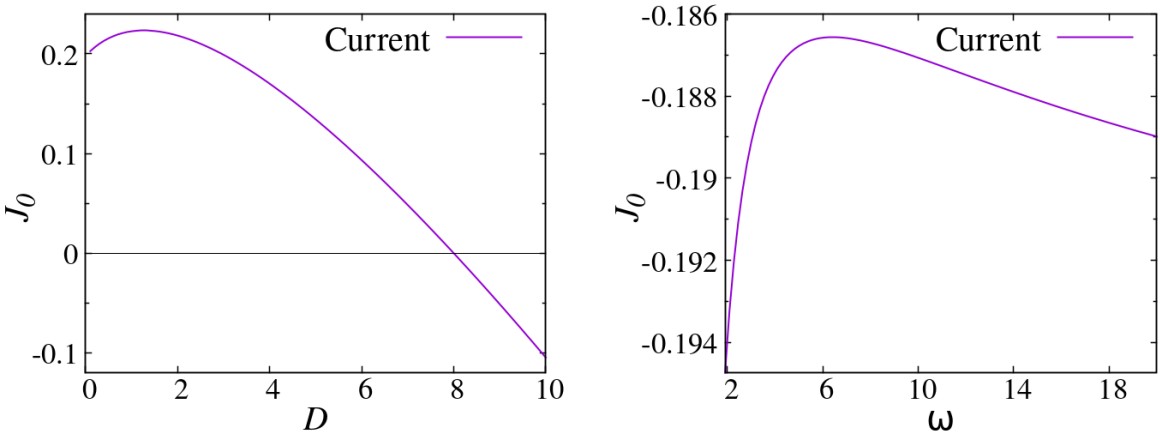

Figure 3: Nonmonotonicity of the steady state current. Left panel: Diffusivity $D$ is tuned. Here $v = 1.0, w = 0.5, L = 200.0, P_0 = Q_0 = 0.002, P_1 = Q_1 = 0.004$. A current reversal also occurs at higher values of $D$. Right panel: Tumble rate $\omega$ is tuned. Here $v = 1.0, D = 1.0, L = 200.0, P_0 = 0.002, P_1 = 0.006, Q_0 = Q_1 = 0.001$. (In both panels the $y$-axis is amplified $10^4$ times for better visibility.)

It is surprising that this elementary model accommodates such remarkable out of equilibrium

---

[2]The probability of finding the particle in the system is, $\int_0^L P(x)\, dx = \frac{P_0+P_1}{2}L$, which is identical to that of a passive particle. This constrains $P_0$ and $P_1$ to $O(L^{-1})$ or less since the total probability cannot exceed 1. We choose the boundary conditions accordingly.

features, some of which are new according to our knowledge. These features might be general in the boundary driven active processes.

## 3.1 Steady state result for zero boundary magnetisation:

For the special case $P_0^+ = P_0^- = \frac{P_0}{2}$ and $P_1^+ = P_1^- = \frac{P_1}{2}$, i.e. $Q_0 = Q_1 = 0$, the solution reduces to a simpler form,

$$P(x) = P_0 + \frac{\Delta P}{L + M \frac{v}{\omega}} \left[ x + l_M \frac{(1 - e^{-\mu L}) - (e^{-\mu x} - e^{-\mu(L-x)})}{1 + e^{-\mu L}} \right], \tag{10}$$

$$Q(x) = \frac{\Delta P}{2 \left( L \frac{\omega}{v} + M \right)} \left[ \frac{e^{-\mu x} + e^{-\mu(L-x)}}{1 + e^{-\mu L}} - 1 \right], \text{ and} \tag{11}$$

$$J_0 = -\frac{D_e}{M \frac{v}{\omega} + L} \Delta P. \tag{12}$$

The equations for the steady state and the boundary conditions in this case are related to the exit probability $E_\pm(x)$ of the particle at the left boundary starting at an initial position $x$ with initial spin $\pm 1$. More details are in Appendix A. It gives a simulation scheme for the steady state. The data with zero boundary magnetisation is shown in Figure 4, which agrees with the theoretical expressions in Eqs. (10)-(11). We however couldn't simulate the steady state with nonzero boundary magnetisations.

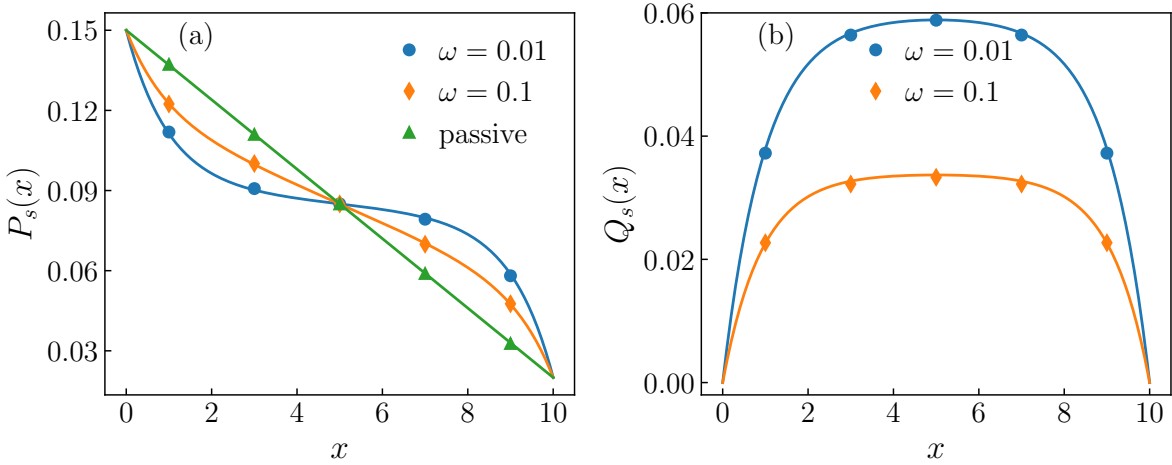

Figure 4: Steady state behaviour for $Q_0 = Q_1 = 0$ — (a): the probability density; (b): magnetisation for different tumble rate $\omega$. Here $L = 10, v = 1.0, D = 1.0$ and the boundary densities are $P_0 = 0.15, P_1 = 0.02$. Points are obtained by simulating the exit probabilities and using Eq. (A.1).

**Large-$L$ limit:** For $L \gg \frac{v}{\omega}, \mu^{-1}$, the particle undergoes many tumble events and we expect the behaviour to be largely diffusive with effective diffusion constant $D_e$. In this limit the steady state results can be quite accurately approximated as [3],

$$J_0 \simeq -D_e \frac{\Delta P}{L + 2 l_M}, \tag{13}$$

$$P(x) \simeq P_0 + \frac{\Delta P}{L + 2 l_M} (x + l_M) - \frac{\Delta P}{L + 2 l_M} l_M (e^{-\mu x} - e^{-\mu(L-x)}), \tag{14}$$

$$Q(x) \simeq -\frac{\Delta P}{L + 2 l_M} \frac{v}{2\omega} (1 - e^{-\mu x} - e^{-\mu(L-x)}). \tag{15}$$

---

[3] The approximation carries a tiny error $\sim O(e^{-\mu L})$.

The expressions suggest that the current and density in the bulk are passive-like with a slightly larger effective system size, whereas the signature of activity shows up at the boundary layer. Nontrivial effect of activity appears in the bulk through $Q(x)$, for which there is no passive analogue and which assumes a constant value for $\mu^{-1} \ll x \ll L$.

## 4 Eigenspectrum and the time-dependent solution

The evolution of the distribution of the RTP, given by the master equation (2a)-(2b), can be rewritten in terms of the state vector $|P(x,t)\rangle = \begin{bmatrix} P_+(x,t) \\ P_-(x,t) \end{bmatrix}$ as,

$$\partial_t |P(x,t)\rangle = \mathcal{L}|P(x,t)\rangle, \tag{16}$$

$\mathcal{L}$ being a linear operator,

$$\mathcal{L} = \begin{bmatrix} D\partial_x^2 - v\partial_x - \omega & \omega \\ \omega & D\partial_x^2 + v\partial_x - \omega \end{bmatrix}, \tag{17}$$

along with the boundary conditions Eq. (3). The solution to Eq. (16) subject to the initial condition gives the full state vector $|P(x,t)\rangle$, which can be written as,

$$|P(x,t)\rangle = |P_{\text{ss}}(x)\rangle + \sum_{n=1}^{\infty} c_n e^{\lambda_n t} |\phi_n(x)\rangle, \tag{18}$$

where $|P_{\text{ss}}(x)\rangle$ is the steady state, and $|\phi_n(x)\rangle \equiv [\phi_n^+(x), \phi_n^-(x)]^T$ are the eigenfunctions of the time evolution operator $\mathcal{L}$. Substituting Eq. (18) in Eq. (16) and using the linear independence of $e^{\lambda_n t}$'s for different $\lambda_n$'s, we find,

$$\mathcal{L}|\phi_n(x)\rangle = \lambda_n |\phi_n(x)\rangle. \tag{19}$$

Here $\lambda_n$'s are the eigenvalues of $\mathcal{L}$ with corresponding eigenfunctions $|\phi_n(x)\rangle$, both yet unknown, satisfying the boundary conditions $|\phi_n(0)\rangle = |\phi_n(L)\rangle = 0$ as we shall argue soon. The coefficients $c_n$ are determined from the initial condition. Let us consider the eigenfunctions $|\phi_n(x)\rangle$ to be of the form,

$$|\phi_n(x)\rangle = \alpha_n e^{k_n x} |A_n\rangle \tag{20}$$

where $|A_n\rangle = [A_n^+, A_n^-]^T$. Using the above, Eq. (19) can be written as,

$$M(k_n)|A_n\rangle = \lambda_n |A_n\rangle \tag{21}$$

with

$$M(k_n) = \begin{bmatrix} Dk_n^2 - vk_n - \omega & \omega \\ \omega & Dk_n^2 + vk_n - \omega \end{bmatrix}. \tag{22}$$

The eigenvalues of $M(k_n)$ gives the 'dispersion relation':

$$\lambda_n = -(\omega - Dk_n^2) \pm \sqrt{\omega^2 + v^2 k_n^2} . \tag{23}$$

To facilitate the subsequent analysis we invert the above relation and express $k_n$ in terms of $\lambda_n$,

$$k_n^2 = \frac{\lambda_n}{D} + \frac{\omega D_e}{D^2} \pm \sqrt{\left(\frac{\lambda_n}{D} + \frac{\omega D_e}{D^2}\right)^2 - \frac{\lambda_n^2 + 2\omega\lambda_n}{D^2}} , \tag{24}$$

Denoting the positive roots for the $k_n$'s corresponding to plus and minus sign by $k_a$ and $k_b$ respectively, Eq. (24) implies that there are four $k_n$ values for each $\lambda_n$,

$$k_n^{(1)} = k_a(\lambda_n), \quad k_n^{(2)} = -k_a(\lambda_n) \tag{25a}$$

$$k_n^{(3)} = k_b(\lambda_n), \quad k_n^{(4)} = -k_b(\lambda_n). \tag{25b}$$

The corresponding eigenvectors of $M(k_n)$ are

$$|A_n^{(i)}\rangle = \begin{bmatrix} a(k_n^{(i)}) \\ 1 \end{bmatrix}, i = 1, ..., 4 \text{ with} \tag{26}$$

$$a(k) = (\lambda + \omega - kv - Dk^2)/\omega. \tag{27}$$

Note that, $a(k_n^{(1)}) \, a(k_n^{(2)}) = a(k_n^{(3)}) \, a(k_n^{(4)}) = 1$. Therefore, each eigenvalue $\lambda_n$ is four-fold degenerate and the corresponding eigenfunction $|\phi_n(x)\rangle$ can be written as a linear combination:

$$|\phi_n(x)\rangle = \sum_{i=1}^{4} \alpha_n^i e^{k_n^{(i)} x} |A_n^i\rangle. \tag{28}$$

The full time-dependent solution for the probability distribution then becomes,

$$|P(x,t)\rangle = |P_{ss}(x)\rangle + \sum_{n=1}^{\infty} c_n e^{\lambda_n t} \sum_{i=1}^{4} \alpha_n^i e^{k_n^{(i)} x} |A_n^i\rangle. \tag{29}$$

The task is to determine $\lambda_n$'s and $\alpha_n^i$'s in terms of the parameters governing the dynamics, viz $v$, $\omega$, $D$ and the system size $L$. Recall the boundary conditions form Eq. (3),

$$|P(0,t)\rangle = \begin{bmatrix} P_0^+/2 \\ P_0^-/2 \end{bmatrix} = |P_{ss}(0)\rangle,$$
$$|P(L,t)\rangle = \begin{bmatrix} P_1^+/2 \\ P_1^-/2 \end{bmatrix} = |P_{ss}(L)\rangle. \tag{30}$$

The above imply that, at $x = 0$, $|P(0,t)\rangle - |P_{ss}(0)\rangle = 0$, and similar at $x = L$. In fact, $|P_{\mathrm{tr}}(x,t)\rangle \equiv |P(x,t)\rangle - |P_{ss}(x)\rangle$ satisfies the masters equation (2a)-(2b) but with the boundary conditions $|P_{\mathrm{tr}}(0,t)\rangle = |P_{\mathrm{tr}}(L,t)\rangle = 0$. Consequently, $\sum_{n=1}^{\infty} c_n e^{\lambda_n t}|\phi_n(0)\rangle = \sum_{n=1}^{\infty} c_n e^{\lambda_n t}|\phi_n(L)\rangle = 0$ at all times. Using the linear independence of $e^{\lambda_n t}$'s once again, we find, for each $n$,

$$|\phi_n(0)\rangle = |\phi_n(L)\rangle = 0. \tag{31}$$

The time-dependent behaviour of the boundary driven case is therefore identical to that of the one-dimensional dynamics with two absorbing ends at $x = 0$ and $L$. Henceforth, we shall omit the steady state. Using the absorbing boundary conditions in Eq.(28), we get the following set of equations for the coefficients $\alpha_n^i$ :

$$x = 0: \quad \sum_{i=1}^{4} \alpha_n^i = 0, \quad \sum_{i=1}^{4} \alpha_n^i a_n^i = 0 \tag{32a}$$

$$x = L: \quad \sum_{i=1}^{4} \alpha_n^i e^{k_n^{(i)} L} = 0, \quad \sum_{i=1}^{4} \alpha_n^i e^{k_n^{(i)} L} a_n^i = 0 \tag{32b}$$

where $a_n^i \equiv a(k_n^{(i)})$. Above equations can be re-written as a matrix equation, $S(k_n^{(i)})|\alpha\rangle = 0$, where $|\alpha\rangle = [\alpha_n^{(1)}, \alpha_n^{(2)}, \alpha_n^{(3)}, \alpha_n^{(4)}]^T$ and

$$S = \begin{bmatrix} 1 & 1 & 1 & 1 \\ a_n^1 & a_n^2 & a_n^3 & a_n^4 \\ e^{k_n^{(1)}L} & e^{k_n^{(2)}L} & e^{k_n^{(3)}L} & e^{k_n^{(4)}L} \\ a_n^1 e^{k_n^{(1)}L} & a_n^2 e^{k_n^{(2)}L} & a_n^3 e^{k_n^{(3)}L} & a_n^4 e^{k_n^{(4)}L} \end{bmatrix} \tag{33}$$

To have a nontrivial solution for $\alpha$'s we must have

$$det[S] = 0, \tag{34}$$

which, upon solving, determines $\lambda_n$'s as a function of system parameters. We observe that, $\lambda = 0, -2\omega$ satisfies the above determinant equation. $\lambda = 0$ stands for the steady state which is just zero for the absorbing boundaries and is discarded. $\lambda = -2\omega$ corresponds to the tumble dynamics.

## 4.1 Symmetries of the model and solution for the spectrum

The determinant equation (34) is a complicated transcendental equation and we cannot find explicit solutions in general. However, the problem can be greatly simplified. Note that, the master equation (16) satisfies a *symmetry*, that is, whenever the spin is reversed ($\sigma \to -\sigma$) along with $x \to L - x$, the equation and the boundary conditions remain invariant. To express it mathematically, let us define an operation $\mathcal{O}_x$:

$$\mathcal{O}_x[P^+(x), P^-(x)]^T = [P^-(L-x), P^+(L-x)]^T, \tag{35}$$

as shown in Figure 5. We find that, $\mathcal{O}_x$ commutes with $\mathcal{L}$, i.e. it is a symmetry operation. It also satisfies $\mathcal{O}_x^2 = I$, implying that its eigenvalues are $o_x = \pm 1$. Consequently, the eigenstates $|\phi_n(x)\rangle$ of the operator $\mathcal{L}$ will also be an eigenstate of $\mathcal{O}_x$ corresponding to an eigenvalue $\pm 1$. In the following, we segregate all the eigenstates $|\phi_n(x)\rangle$ into two symmetry sectors labelled by the values of $o_x$, and determine the spectrum for each of the sectors.

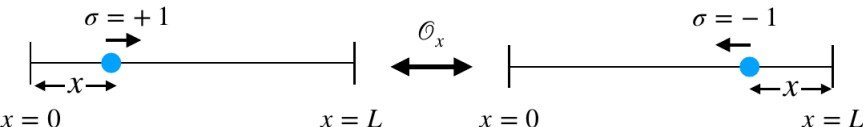

Figure 5: The symmetry of the dynamics. The master equations and the boundary conditions remain invariant under $\sigma \to -\sigma$ along with $x \to L - x$.

**spectrum for $o_x = 1$ (even sector):**
Here, $\phi_n^+(x) = \phi_n^-(L-x)$. Using Eq. (28) and the values of $k_n$'s from Eq. (25), we find,

$$\alpha^{(1)} a^{(1)} e^{k_a x} + \alpha^{(3)} a^{(3)} e^{k_b x} + \alpha^{(2)} a^{(2)} e^{-k_a x} + \alpha^{(4)} a^{(4)} e^{-k_b x} \tag{36}$$

$$= \alpha^{(1)} e^{k_a (L-x)} + \alpha^{(3)} e^{k_b (L-x)} + \alpha^{(2)} e^{-k_a (L-x)} + \alpha^{(4)} e^{-k_b (L-x)}. \tag{37}$$

We dropped the suffix $n$ for convenience. Comparing the coefficients of each independent spatial modes $e^{k^{(i)}x}$, we obtain,

$$\alpha^{(1)} a^{(1)} = \alpha^{(2)} e^{-k_a L} \tag{38}$$

$$\alpha^{(3)} a^{(3)} = \alpha^{(4)} e^{-k_b L} \tag{39}$$

Using the above expressions in Eqs. (32) for the coefficients $\alpha^{(i)}$, we obtain,

$$\frac{\alpha^{(1)}}{\alpha^{(3)}} = -\frac{1 + a^{(3)}e^{k_b L}}{1 + a^{(1)}e^{k_a L}} = -\frac{a^{(3)} + e^{k_b L}}{a^{(1)} + e^{k_a L}}. \tag{40}$$

Rearranging the last equality we get,

$$\frac{e^{-k_b L} - e^{-k_a L}}{1 - e^{-k_a L}e^{-k_b L}} = \frac{a^{(3)} - a^{(1)}}{a^{(3)}a^{(1)} - 1}. \tag{41}$$

Here $k_a, k_b, a^{(1)}, a^{(3)}$ are all functions of $\lambda$. By solving the above expression, we can in principle find the $\lambda$'s in this sector.

**spectrum for $o_x = -1$ (odd sector):**
Here, $\phi_n^+(x) = -\phi_n^-(L - x)$. Following the same procedure, we obtain the equation satisfied by the spectrum in this sector as,

$$\frac{e^{-k_b L} - e^{-k_a L}}{1 - e^{-k_a L}e^{-k_b L}} = -\frac{a^{(3)} - a^{(1)}}{a^{(3)}a^{(1)} - 1}. \tag{42}$$

Although the equations (41)-(42) are exact and much simpler compared to Eq. (34), these are not yet exactly solvable for $\lambda$ in general. However, we earlier mentioned that $\lambda = 0$ and $\lambda = -2\,\omega$ are eigenvalues of the evolution operator since these satisfy Eq. (34). Here we explicitly find that $\lambda = 0$ satisfies Eq. (42) and therefor belongs to the odd sector (which, as argued earlier, should be discarded for these set of solutions). Whereas $\lambda = -2\,\omega$ satisfies Eq. (41) and belongs to the even sector. We find analytical expressions for the 'band' of eigenvalues close to 0 and $-2\,\omega$ in the large $L$ limit and the corresponding eigenfunctions in the two symmetry sectors. In the following, we demonstrate the results and discuss the relaxation behaviour.

### 4.1.1 Spectrum in the large-$L$ limit

**Eigenvalues near $\lambda = 0$ and relaxation:** In the limit of large system size, we expect that the eigenvalues are very close to zero and therefore from Eq. (24), we get,

$$k_a \approx \frac{\sqrt{2\omega D_e}}{D}(= \mu), \;\; k_b \approx \sqrt{\lambda/D_e}, \tag{43}$$

The finite value of $k_a$ renders $e^{-k_a L} \approx 0$ in Eqs. (41)-(42). Further, from Eq. (27) one can find that, $a^{(3)} \approx 1$, $a^{(1)} \approx -\frac{1 + v/\sqrt{2\omega D_e}}{1 - v/\sqrt{2\omega D_e}}$, and the ratio in the rhs of the Eqs. (41)-(42) takes the value, $\frac{a^{(3)} - a^{(1)}}{a^{(3)}a^{(1)} - 1} \approx -1$. Consequently, we have, for $o_x = \pm 1$,

$$e^{k_b L} \approx \mp 1 \quad \Rightarrow \quad e^{\sqrt{\frac{\lambda}{D_e}}L} \approx \mp 1 \tag{44}$$

The solutions are,

$$\lambda_n \approx -\frac{(2n - 1)^2\pi^2 D_e}{L^2}, \;\; n > 0 \text{ (for } o_x = 1), \tag{45a}$$

$$\lambda_{n'} \approx -\frac{(2n')^2\pi^2 D_e}{L^2}, \;\; n' > 0 \quad \text{(for } o_x = -1). \tag{45b}$$

These are the alternate symmetric and antisymmetric bands near $\lambda = 0$. We find that the relaxation rate, that is given by the eigenvalue with real part closest to zero, lies in the symmetric sector and in the large $L$ limit it is given by,

$$|\lambda_1| = \frac{\pi^2 D_e}{L^2}. \tag{46}$$

Alternatively, the relaxation time is, $\tau_L = |\lambda_1|^{-1} = \frac{L^2}{\pi^2 D_e}$. This suggests that, for large systems the relaxation behaviour resembles that of a Brownian particle with an effective diffusion rate $D_e$. Note that, the solution given in Eq. (45) for the band of eigenvalues are valid as long as $|\lambda_n| \ll 2\omega$, or, $n \ll \sqrt{\tau_L/\tau_t}$, where $\tau_t = (2\omega)^{-1}$ is the relaxation rate of the tumble dynamics.

**Correction to the leading behaviour:** To evaluate the subleading behaviour we need to keep the first correction terms in $\lambda$ in Eqs. (41)-(42). Since $k_a$ is finite, in the large $L$ limit we shall neglect $e^{-k_a L}$ and therefore the l.h.s. in both the equations is $e^{-k_b L} \approx e^{-L\sqrt{\lambda/D_e}}$. On the other hand, for small $\lambda$, $a^{(3)} \approx 1 - \frac{v}{\omega\sqrt{D_e}}\sqrt{\lambda}$, whereas $a^{(1)} \approx -\frac{1+v/\sqrt{2\omega D_e}}{1-v/\sqrt{2\omega D_e}} + O(\lambda)$. This implies that the corrections in the rhs of Eqs. (41)-(42) are of $O(\sqrt{\lambda})$, and the equations to the first subleading order reads,

$$e^{-L\sqrt{\lambda/D_e}} \approx \mp \left(1 + \frac{v^2}{\omega D_e}\sqrt{\frac{\lambda}{2\omega}}\right), \quad \text{for } o_x = \pm 1 \text{ respectively.} \tag{47}$$

One can solve the above equation perturbatively by considering $\lambda_n = \lambda_n^{(0)} + \lambda_n^{(1)}$, where $\lambda_n^{(0)} = -\frac{n^2\pi^2 D_e}{L^2}$ are the solution at the leading order in $L$ and $\lambda_n^{(1)} \ll \lambda_n^{(0)}$ are the corrections. We want to find the leading $L$-dependence of $\lambda_n^{(1)}$. Putting $\lambda_n = \lambda_n^{(0)} + \lambda_n^{(1)}$ in Eq. (47), we get,

$$e^{-L\sqrt{\frac{\lambda_n^{(0)}+\lambda_n^{(1)}}{D_e}}} \approx \mp \left(1 + \frac{v^2}{\omega D_e}\sqrt{\frac{\lambda_n^{(0)} + \lambda_n^{(1)}}{2\omega}}\right). \tag{48}$$

To the first subleading order the left hand side of the Eq. (48) can be written as,

$$e^{-k_b L} \approx e^{-L\sqrt{\frac{\lambda_n^{(0)}+\lambda_n^{(1)}}{D_e}}} = e^{-L\sqrt{\frac{\lambda_n^{(0)}}{D_e}}\left(1+\frac{\lambda_n^{(1)}}{\lambda_n^{(0)}}\right)^{1/2}} \approx e^{-L\sqrt{\frac{\lambda_n^{(0)}}{D_e}}} e^{-L\sqrt{\frac{\lambda_n^{(0)}}{D_e}}\left(\frac{\lambda_n^{(1)}}{2\lambda_n^{(0)}}\right)} \tag{49}$$

Now, using in the above the expression for $\lambda_n^{(0)}$ from Eq. (45), the l.h.s. of Eq. (48) is obtained as,

$$e^{-k_b L} \approx \mp 1 \times \exp\left(-i\frac{n\pi}{2}\frac{\lambda_n^{(1)}}{\lambda_n^{(0)}}\right) \approx \mp 1 \pm i\frac{n\pi}{2}\frac{\lambda_n^{(1)}}{\lambda_n^{(0)}}, \tag{50}$$

Similarly, we find the right hand side of Eq. (48) to the first subleading order,

$$\mp \left(1 + \frac{v^2}{\omega D_e}\sqrt{\frac{\lambda_n^{(0)} + \lambda_n^{(1)}}{2\omega}}\right) \approx \mp \left(1 + \frac{v^2}{\omega D_e}\sqrt{\frac{\lambda_n^{(0)}}{2\omega}}\right) = \mp 1 \mp i\frac{n\pi}{L}\frac{v^2}{\omega\sqrt{2\omega D_e}}.$$

Then, equating the l.h.s. and r.h.s. of Eq. (48), we get,

$$\lambda_n^{(1)} \approx -\frac{2v^2}{\omega\sqrt{2\omega D_e}}\frac{\lambda_n^{(0)}}{L}, \tag{51}$$

and consequently, the band near $\lambda = 0$ takes the form,

$$\lambda_n = \lambda_n^{(0)} + \lambda_n^{(1)} \approx \lambda_n^{(0)}\left(1 - \frac{2v^2}{\omega\sqrt{2\omega D_e}}\frac{1}{L}\right). \tag{52}$$

**Eigenvalues near $\lambda = -2\omega$:** We have already mentioned that $-2\omega$ is an eigenvalue of the evolution operator that belongs to the symmetric sector. To calculate the band of eigenvalues around it, let us consider $\lambda = -2\omega + \varepsilon$, where $\varepsilon$ is presumably small in the large $L$ limit. In this limit,

$$k_a \approx \sqrt{\frac{v^2}{D^2} + \frac{\lambda}{D}}, \ k_b \approx \sqrt{\varepsilon/D}$$

.

$$a^{(1)} = 1 - \frac{v^2}{\omega D} - \frac{k_a v}{\omega}, \ a^{(3)} = -1 - \frac{k_b v}{\omega}.$$

Since $\varepsilon \to 0$, $a^{(3)} \approx -1$. In the large $L$-limit, $e^{-k_a L} \to 0$. Therefore, Eqs (42)-(43) reduce to

$$e^{-k_b L} = \pm 1 \ \text{ for } o_x = \pm 1. \tag{53}$$

The solutions are,

$$\varepsilon_n \approx -\frac{(2n)^2 \pi^2 D}{L^2}, \ n \geq 0 \quad (\text{for } o_x = 1), \tag{54a}$$

$$\varepsilon_{n'} \approx -\frac{(2n'-1)^2 \pi^2 D}{L^2}, \ n' > 0 \ (\text{for } o_x = -1), \tag{54b}$$

and therefore $\lambda_n = -2\omega + \varepsilon_n$. The corrections to the above expressions can be calculated, which is not shown in the present paper.

## 4.2  Crossover behaviour of the relaxation time

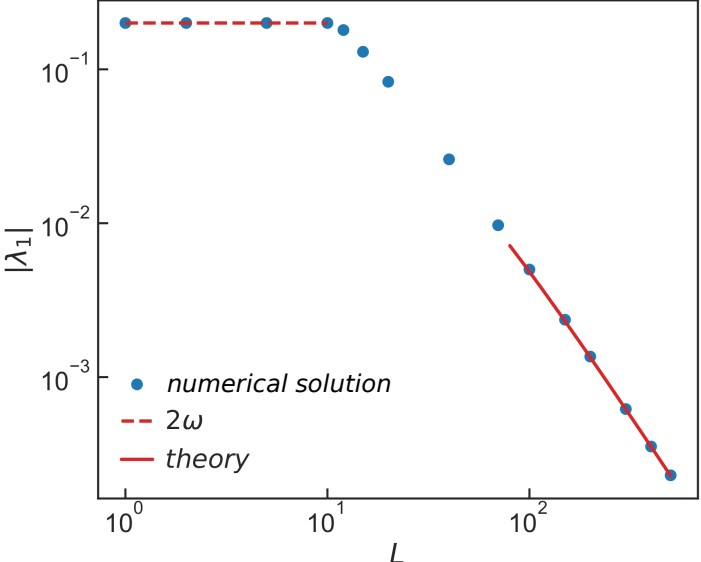

Figure 6: Crossover behaviour of the relaxation rate for different system sizes. Here, $D = 1.0, v = 1.0, \omega = 0.1$. Points are obtained by numerically solving Eq. (41) (symmetric sector) and identifying the solution closest to zero using Mathematica. The crossover to $L$-dependent relaxation is seen to occur for system sizes very close to $L_p = \frac{v}{\omega} = 10.0$. The large $L$ behaviour is consistent with Eq. (52) for $n = 1$.

For passive particles the relaxation time because of diffusion is proportional to $L^2/D$. RTPs on the other hand also carries a tumble dynamics which relaxes in a timescale $\tau_t \sim (2\omega)^{-1}$. When the system size is much larger than the persistent length $L_c \sim v/\omega$, several tumble events

occur before reaching the steady state and the relaxation dynamics becomes effectively passive-like. In this regime the bands are also well-separated. As $L$ is reduced, the corrections due to activity (Eq. (52)) become significant and a deviation from diffusive behaviour shows up. When the $L \sim L_c$, the two bands would mix and for $L < L_c$ they cross, implying that the diffusive dynamics now relaxes faster compared to the tumble dynamics and $\lambda = -2\omega$ determines the relaxation of the system. This behaviour is shown in Figure 6. We expect this to be a generic feature of finite-sized active systems, and more generally of processes with coloured noise.

## 4.3 Eigenfunctions and the large-time distribution

Once we have the spectrum, we can completely solve for the coefficients $\alpha_n^i, i = (1, \ldots, 4)$. For $o_x = \pm 1$, the coefficients are related as follows:

$$\alpha^{(1)} = -\alpha^{(3)} \frac{a^{(3)} \pm e^{k_b L}}{a^{(1)} \pm e^{k_a L}}, \tag{55}$$

$$\alpha^{(2)} = \pm \alpha^{(1)} a^{(1)} e^{k_a L}, \tag{56}$$

$$\alpha^{(4)} = \pm \alpha^{(3)} a^{(3)} e^{k_b L}, \tag{57}$$

and $\alpha^{(3)}$ is left arbitrary at this point. After incorporating the coefficients in Eq. (28), the expression of the eigenfunctions $|\phi_n(x)\rangle$ becomes,

$$|\phi_n(x)\rangle = \alpha_n^{(3)} \left[ \frac{a^{(3)} \pm e^{k_b L}}{a^{(1)} \pm e^{k_a L}} \left( e^{k_a x} \begin{bmatrix} a^{(1)} \\ 1 \end{bmatrix} \pm e^{k_a(L-x)} \begin{bmatrix} 1 \\ a^{(1)} \end{bmatrix} \right) - \left( e^{k_b x} \begin{bmatrix} a^{(3)} \\ 1 \end{bmatrix} \pm e^{k_b(L-x)} \begin{bmatrix} 1 \\ a^{(3)} \end{bmatrix} \right) \right]. \tag{58}$$

The set of equations (55)-(57) can be simplified and the coefficients can be determined order by order in the large $L$ limit. We shall present results for the band near $\lambda = 0$ and keep the terms which are $\sim O(L^{-1})$.

To this order we find, for $o_x = \pm 1$, $e^{k_b L} \approx \mp (1 - \frac{n\pi}{L} \frac{v^2}{\omega \sqrt{2\omega D_e}})$ using Eqs. (50)-(51), and $\alpha^{(1)} = -\alpha^{(3)} \frac{a^{(3)} \pm e^{k_b L}}{a^{(1)} \pm e^{k_a L}} \approx \mp \alpha^{(3)} [a^{(3)} \pm e^{k_b L}] e^{-k_a L}$ from Eq. (55). In Eq. (27), keeping the first subleading term in the expression for $a^{(3)}$ and using Eq. (52) to determine $k_b = \sqrt{\lambda/D_e}$ to $O(L^{-1})$, we find, $a^{(3)} \approx 1 - vk_b/\omega \approx 1 - i\frac{v}{\omega}\frac{n\pi}{L}$. Putting these together in Eq. (55) we obtain,

$$\alpha^{(1)} = \pm i\, \alpha^{(3)} \frac{n\pi}{L} \frac{v}{\omega} \left(1 - \frac{v}{\sqrt{2\omega D_e}}\right) e^{-k_a L}, \text{ for } n = \text{odd/even}. \tag{59}$$

Using the above in Eq. (56) we find,

$$\alpha^{(2)} = i\, \alpha^{(3)} \frac{n\pi}{L} a^{(1)} \frac{v}{\omega} \left(1 - \frac{v}{\sqrt{2\omega D_e}}\right); \tag{60}$$

and finally, from Eq. (57)

$$\alpha^{(4)} = -\alpha^{(3)} a^{(3)} \left(1 - i\frac{n\pi}{L} \frac{v^2}{\omega\sqrt{2\omega D_e}}\right). \tag{61}$$

Using the coefficients in Eq. (29), we find, at large times $t \gg \omega^{-1}$,

$$\begin{aligned} |P_{\text{tr}}(x,t)\rangle &\approx \sum_{n=1}^{\infty} c_n e^{\lambda_n t} \alpha_n^{(3)} \left( 2i \sin\frac{n\pi x}{L} \begin{bmatrix} 1 \\ 1 \end{bmatrix} - i\frac{n\pi v}{\omega L} \cos\frac{n\pi x}{L} \begin{bmatrix} 1 \\ -1 \end{bmatrix} \right. \\ &\quad + \frac{n\pi v}{\omega L} \left\{ \left(1 + \frac{v}{\sqrt{2\omega D_e}}\right) \sin\frac{n\pi x}{L} + i\frac{v}{\sqrt{2\omega D_e}}\left(1 - \frac{2x}{L}\right)\cos\frac{n\pi x}{L} \right\} \begin{bmatrix} 1 \\ 1 \end{bmatrix} \\ &\quad \left. + i\frac{n\pi v}{\omega L}\left(1 - \frac{v}{\sqrt{2\omega D_e}}\right) \left\{ (-1)^{n-1} e^{-k_a(L-x)} \begin{bmatrix} a_n^{(1)} \\ 1 \end{bmatrix} + e^{-k_a x} \begin{bmatrix} 1 \\ a_n^{(1)} \end{bmatrix} \right\} \right), \end{aligned} \tag{62}$$

with $\lambda_n \approx -\frac{n^2\pi^2 D_e}{L^2}$, $n > 0$; here we have used $a_n^{(1)} a_n^{(2)} = 1$. Each term of the summation in Eq. (62) is an eigenfunction evaluated to $O(L^{-1})$. In the following we shall use $\alpha_n$ instead of $\alpha_n^{(3)}$. To the *leading order*, each of the eigenfunctions is simply proportional to $\sin \frac{n\pi x}{L}$ and the probability density takes a passive-like form with an effective diffusion constant $D_e$:

$$\lim_{t\to\infty} \lim_{L\to\infty} P(x,t) \approx \sum_{n=1}^{\infty} c_n \alpha_n \, e^{\lambda_n t} 4\, i\, \sin \frac{n\pi x}{L} \equiv P_{\text{passive}}(x,t)|_{D\to D_e}, \tag{63}$$

while the magnetisation vanishes. In particular, the relaxation mode is given by $\phi_1(x) \propto \sin \frac{\pi x}{L}$. Note that the *active* contributions in excess to the effective passivelike contributions, proportional to the active speed $v$, appear at $O(L^{-1})$ in each term of the distribution $P_{\text{tr}}(x,t) = [1 \ 1]\,|P_{\text{tr}}(x,t)\rangle$ and magnetization $Q_{\text{tr}}(x,t) = [1 \ -1]\,|P_{\text{tr}}(x,t)\rangle$. Thus at very large times when only very few $n$'s contribute, the active part in the distribution and the magnetisation appears only at the subleading order. However, this is not true in the large but intermediate times when larger $n$'s also contribute.

To determine the full distribution we need to evaluate the $n-$dependent constants $c_n$ and $\alpha_n$. Let $\langle \psi_n(x)|$ be the left eigenvector of the evolution operator $\mathcal{L}$ corresponding to eigenvalue $\lambda_n$. Then the orthonormality of the $\langle \psi_n(x)|$'s and $|\phi_n(x)\rangle$'s give us $\alpha_n$. The $c_n$ is subsequently determined from the initial condition: $c_n = \int dx\, \langle \psi_n(x)|P(x,0)\rangle$. We shall consider a fixed initial condition, $|P(x,0)\rangle = \delta(x - x_0) \begin{bmatrix} s_+ \\ s_- \end{bmatrix}$, and find the propagator at large times. Here $s_+, s_-$ are the probabilities of the initial spin to be plus and minus respectively: $s_+ + s_- = 1$, and $m_0 = s_+ - s_-$ is the initial magnetisation. The detailed expressions for $\langle \psi_n(x)|$, $\alpha_n$ and $c_n$'s are given in the Appendix B. Incorporating these, we obtain the expression of the state to $O(L^{-1})$ for times $t \gg \omega^{-1}$,

$$
\begin{aligned}
|P_{\text{tr}}(x,t)\rangle &= \frac{1}{2L} \sum_{n=1}^{\infty} e^{\lambda_n t} \Bigg( 2 \left\{ 1 - \frac{v^2}{L\omega\sqrt{2\omega D_e}} \right\} \sin \frac{n\pi x_0}{L} \sin \frac{n\pi x}{L} \begin{bmatrix} 1 \\ 1 \end{bmatrix} \\
&\quad + \frac{n\pi\, v^2}{L\omega\sqrt{2\omega D_e}} \left\{ (1 - \frac{2x_0}{L}) \cos \frac{n\pi x_0}{L} \sin \frac{n\pi x}{L} + (1 - \frac{2x}{L}) \cos \frac{n\pi x}{L} \sin \frac{n\pi x_0}{L} \right\} \begin{bmatrix} 1 \\ 1 \end{bmatrix} \\
&\quad - \frac{n\pi v}{L\omega} \left\{ (\frac{v}{\sqrt{2\omega D_e}} + m_0) e^{-k_a x_0} - (-1)^n (\frac{v}{\sqrt{2\omega D_e}} - m_0) e^{-k_a(L-x_0)} \right\} \sin \frac{n\pi x}{L} \begin{bmatrix} 1 \\ 1 \end{bmatrix} \\
&\quad + \frac{n\pi v}{L\omega} (1 - \frac{v}{\sqrt{2\omega D_e}}) \sin \frac{n\pi x_0}{L} \left\{ e^{-k_a x} \begin{bmatrix} 1 \\ a^{(1)} \end{bmatrix} - (-1)^n e^{-k_a(L-x)} \begin{bmatrix} a^{(1)} \\ 1 \end{bmatrix} \right\} \\
&\quad + m_0 \frac{n\pi v}{L\omega} \cos \frac{n\pi x_0}{L} \sin \frac{n\pi x}{L} \begin{bmatrix} 1 \\ 1 \end{bmatrix} - \frac{n\pi v}{L\omega} \sin \frac{n\pi x_0}{L} \cos \frac{n\pi x}{L} \begin{bmatrix} 1 \\ -1 \end{bmatrix} \Bigg). \tag{64}
\end{aligned}
$$

At relaxation time scales ($t \gtrsim L^2/D_e$), only the small $n$ terms ($n = 1, 2$ etc.) survive. Here the dominating contribution comes from the first term of Eq. (64) which is just the effective passive expression, while the active contribution occurs as subleading corrections. However, at intermediate times $\omega^{-1} \ll t \ll L^2/D_e$, all the $n$'s contribute and it is not obvious whether the active contributions constitute only a correction to the effective passive behaviour. For this we aim to evaluate the summations in a closed form. Generally this is not doable for the summations at hand, but we can make progress for large values of $L$ using the Poisson summation formula and the symmetries of the distribution. The details and a closed form expression of $|P_{\text{tr}}(x,t)\rangle$ is given in Appendix B. For finite $x_0 \ll L$ the system is well approximated by a semi-infinite line, for which we discuss the results below.

## 4.4 Semi-infinite line with absorbing barrier at $x = 0$:

On the semi-infinite geometry the summations in the Eq. (64) can be converted to integrals and the state vector at large times take the form:

$$
\begin{aligned}
|P_{\text{tr}}^{\infty}(x,t)\rangle &= \frac{1}{2}\left( \frac{1}{\sqrt{4\pi D_e t}} \left\{ e^{-\frac{(x-x_0)^2}{4D_e t}} - e^{-\frac{(x+x_0)^2}{4D_e t}} \right\} \begin{bmatrix} 1 \\ 1 \end{bmatrix} \right.\\
&\quad + \frac{m_0 v}{\omega} \frac{\pi}{(4\pi D_e t)^{3/2}} \left\{ (x-x_0)\, e^{-\frac{(x-x_0)^2}{4D_e t}} + (x+x_0)\, e^{-\frac{(x+x_0)^2}{4D_e t}} \right\} \begin{bmatrix} 1 \\ 1 \end{bmatrix}\\
&\quad + \frac{v}{\omega} \frac{2\pi}{(4\pi D_e t)^{3/2}} \left\{ \frac{v}{\sqrt{2\omega D_e}}(x+x_0)\, e^{-\frac{(x+x_0)^2}{4D_e t}} - \left(\frac{v}{\sqrt{2\omega D_e}} + m_0\right) x\, e^{-k_a x_0}\, e^{-\frac{x^2}{4D_e t}} \right\} \begin{bmatrix} 1 \\ 1 \end{bmatrix}\\
&\quad + \frac{v}{\omega}\left(1 - \frac{v}{\sqrt{2\omega D_e}}\right) \frac{2\pi}{(4\pi D_e t)^{3/2}} x_0\, e^{-k_a x}\, e^{-\frac{x_0^2}{4D_e t}} \begin{bmatrix} 1 \\ a^{(1)} \end{bmatrix}\\
&\quad \left. + \frac{v}{\omega} \frac{\pi}{(4\pi D_e t)^{3/2}} \left\{ (x-x_0)\, e^{-\frac{(x-x_0)^2}{4D_e t}} - (x+x_0)\, e^{-\frac{(x+x_0)^2}{4D_e t}} \right\} \begin{bmatrix} 1 \\ -1 \end{bmatrix} \right). \qquad (65)
\end{aligned}
$$

The same expression is obtained by taking the limit $L \gg x, x_0$ in Eq. (B.7). All the terms in the above expression are $\sim O(t^{-3/2})$, and therefore 'passive' as well as the 'active' contributions are equally important in determining the large time behaviour in the semi infinite domain. From Eq. (65) we find the distribution and magnetisation profile by respectively adding and subtracting the rows:

$$
\begin{aligned}
P_{\text{tr}}^{\infty}(x,t) &= \frac{1}{\sqrt{4\pi D_e t}} \left\{ e^{-\frac{(x-x_0)^2}{4D_e t}} - e^{-\frac{(x+x_0)^2}{4D_e t}} \right\}\\
&\quad + \frac{v^2}{\omega\sqrt{2\omega D_e}} \frac{2\pi}{(4\pi D_e t)^{3/2}}(x+x_0)\, e^{-\frac{(x+x_0)^2}{4D_e t}}\\
&\quad + \frac{m_0 v}{\omega} \frac{\pi}{(4\pi D_e t)^{3/2}} \left\{ (x-x_0)\, e^{-\frac{(x-x_0)^2}{4D_e t}} + (x+x_0)\, e^{-\frac{(x+x_0)^2}{4D_e t}} \right\}\\
&\quad - \frac{v}{\omega} \frac{2\pi}{(4\pi D_e t)^{3/2}} \left\{ \left(\frac{v}{\sqrt{2\omega D_e}} + m_0\right) x\, e^{-k_a x_0}\, e^{-\frac{x^2}{4D_e t}} + \frac{v}{\sqrt{2\omega D_e}} x_0\, e^{-k_a x}\, e^{-\frac{x_0^2}{4D_e t}} \right\}, \quad (66)
\end{aligned}
$$

and,

$$
\begin{aligned}
Q_{\text{tr}}^{\infty}(x,t) &= \frac{v}{\omega} \frac{\pi}{(4\pi D_e t)^{3/2}} \left\{ (x-x_0)\, e^{-\frac{(x-x_0)^2}{4D_e t}} - (x+x_0)\, e^{-\frac{(x+x_0)^2}{4D_e t}} \right\}\\
&\quad + \frac{v}{\omega} \frac{2\pi}{(4\pi D_e t)^{3/2}} x_0\, e^{-k_a x}\, e^{-\frac{x_0^2}{4D_e t}}. \qquad (67)
\end{aligned}
$$

In Figure 7 we have shown the distribution and magnetisation obtained by simulating the RTP dynamics on the semi-infinite line and its comparison with the theoretical predictions. The profiles carry an interesting structure. In each of Eqs. (66)-(67), the term that falls exponentially in distance represents the kinetic boundary layer with a 'skin depth' $k_a^{-1}$, identical to that in the steady state. The other terms form the 'scaling' part of the profile. In Eq. (66) the last but one term falls rapidly if the initial position exceeds the skin depth.

In Eq. (66) we can immediately identify the *passive* and *active* contributions to the distribution. It is useful to look at the relative magnitude of these two parts. We first note that, at space-time scales such that $t \gg \{\frac{x_0^2}{D_e}, \frac{x^2}{D_e}\}$, $P_{\text{passive}}^{\infty}(x,t) = \frac{1}{\sqrt{4\pi D_e t}}\left\{ e^{-\frac{(x-x_0)^2}{4D_e t}} - e^{-\frac{(x+x_0)^2}{4D_e t}} \right\} \approx \frac{x x_0}{2\sqrt{\pi}\,(D_e t)^{3/2}}$ and from Eq. (66), $P_{\text{active}}^{\infty} = P_{\text{tr}} - P_{\text{passive}}^{\infty} \sim O(t^{-3/2})$, i.e. both occur at the same order. The relative weight of active and passive parts is,

$$
\tilde{R}(x_0, x) = \lim_{t \gg \frac{x_0^2}{D_e}, \frac{x^2}{D_e}} \frac{P_{\text{active}}^{\infty}}{P_{\text{passive}}^{\infty}} = \frac{v}{2\omega}\left[ \left(m_0 + \frac{v}{\sqrt{2\omega D_e}}\right)\frac{1 - e^{-k_a x_0}}{x_0} + \frac{v}{\sqrt{2\omega D_e}}\frac{1 - e^{-k_a x}}{x} \right], \quad (68)
$$

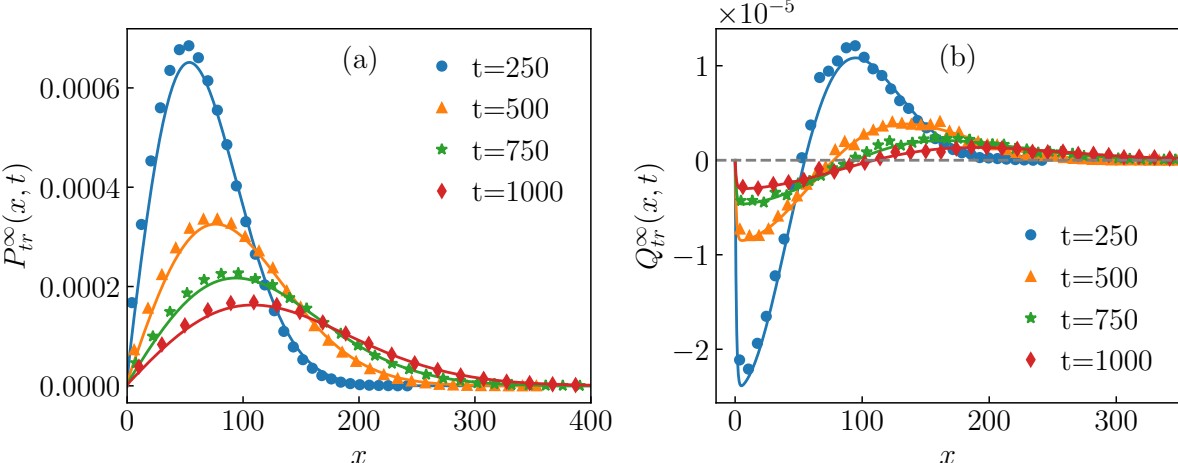

Figure 7: (a) Probability density at different times. Solid lines are the theoretical expression (Eq. (66)). (b) The magnetisation profile at different times. Solid lines are from the expression in Eq. (67). Here $x_0 = 1$, $m_0 = 0$, $\omega = 0.1$, $v = 1$, $D = 1$.

which is finite throughout the system for any finite $x_0$, and can actually attain dominant role for highly persistent (small $\omega$ or large $v$) run and tumble motion. The effect of activity becomes more prominent when the particle is in the 'boundary layer' $x \lesssim k_a^{-1}$. Beyond that this part decays slowly as $x^{-1}$, and asymptotically the contribution from initial position $x_0$ only remains. In fact, it can be shown from Eq. (66) that, when both $x$ and $t$ is very large such that $u = x/\sqrt{D_e t}$ is finite, both the active and passive contributions are $\sim (D_e t)^{-1}$, and their ratio becomes,

$$R(x_0) = \lim_{\{x,t\to\infty, \frac{x}{\sqrt{D_e t}} = u\}} \frac{P^\infty_{\text{active}}}{P^\infty_{\text{passive}}} = \frac{v}{2\,\omega}\Big(m_0 + \frac{v}{\sqrt{2\,\omega\,D_e}}\Big)\frac{1 - e^{-k_a x_0}}{x_0}. \tag{69}$$

The nature of $R(x_0)$ is shown in Figure 8. To see the impact of the active contribution, consider

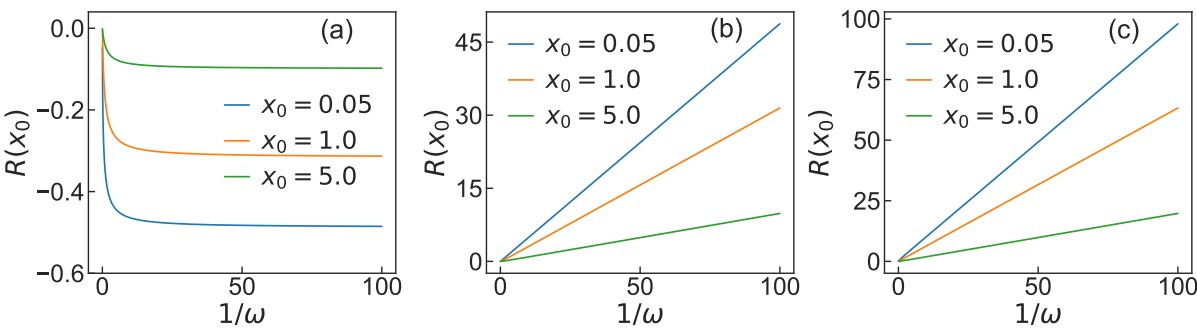

Figure 8: $R(x_0)$ from Eq. (69) as a function of $1/\omega$ for (a) $m_0 = -1$, (b) $m_0 = 0$, (c) $m_0 = 1$ for different values of $x_0$. Here $v = D = 1$.

a highly persistent (small $\omega$) particle starting at some $x_0 \lesssim k_a^{-1} = \frac{D}{\sqrt{2\,\omega\,D_e}} \approx \frac{D}{v}$. In this case $R(x_0) \approx \frac{v}{2\,\omega}\big(m_0 + \frac{v}{\sqrt{2\,\omega\,D_e}}\big)k_a \approx \frac{v^2}{\omega D}s_+ - \frac{1}{2}$: if the probability $s_+$ of the initial spin to be positive is finite, the active to passive ratio is very high and the distribution in the bulk is almost completely dominated by the active part. This includes the $m_0 = 0$ case which is closest to a passive-like initial state. This also tells that a perturbation in the active case will spread very differently from its passive counterpart; moreover, an active 'intrusion' ($x_0 \approx 0$) will in general stay longer and penetrate further into the system compared to a passive one of same effective diffusivity.

### 4.4.1 Discussion: thermal noise and the late time behaviour of active particles

This everlasting and often dominant influence of activity to the distribution is partly related to the initial penetration of the particles before they start tumbling. But that effect is strongly modulated in the presence of thermal noise. The nontrivial effect of diffusion is reflected in the flux at $x = 0$:

$$
\begin{aligned}
J(0,t) &= \left[ -D\frac{\partial P(x,t)}{\partial x} + v\, Q(x,t) \right]_{x=0} \\
&= -\frac{1}{2\sqrt{\pi}(D_e t)^{\frac{3}{2}}} \left[ D_e\, x_0 + \frac{Dv}{2\,\omega}(m_0 + \frac{v}{\sqrt{2\,\omega\, D_e}})(1 - e^{-k_a x_0}) \right].
\end{aligned}
\tag{70}
$$

For $D = 0$, the terms in the square bracket reduce to $D_e\, x_0$, similar to passive particles; and this flux is due to the particles with instantaneous negative spin. For nonzero but small $D$, $e^{-k_a x_0}$ is still negligible and the flux gains a contribution that couples thermal noise and activity: $\frac{Dv}{2\omega}(m_0 + 1 - \frac{\omega D}{v^2}) = D(\frac{v}{\omega}s_+ - \frac{D}{2v})$. If $s_+ > 0$ the change is proportional to $\frac{Dv}{\omega}$ which is positive and the particles with *initial positive spin* largely contribute to an additional flux at the origin. This is consistent with the picture that at short times the particles with initial positive spin move away from the origin, which makes them available in the system to contribute to the noise driven flux at large times. There is also a spin-independent reduction of the flux amounting to $-\frac{D^2}{2v}$, which is much smaller but gives the dominating correction for $s_+ = 0$. It pertains to the diffusive spread $\sim O(\frac{D}{v})$ over the persistent motion.

The magnetisation also gives important insights to the dynamics at large times. As expected, $Q_{\mathrm{tr}}(x,t)$ vanishes in the passive case ($v = 0$). It is interesting to note that the magnetisation profile in Eq. (67) is independent of the initial magnetisation $m_0$, although the probability density strongly depends on it. At large times the total magnetisation in the system is given by,

$$
Q^\infty(t) = \int_0^\infty Q_{\mathrm{tr}}^\infty(x,t)\, dx = \frac{v\, D}{\omega\sqrt{2\omega D_e}} \frac{2\pi}{(4\pi D_e t)^{3/2}} x_0\, e^{-\frac{x_0^2}{4 D_e t}},
\tag{71}
$$

which is positive at all times and undergoes a slow algebraic decay. It is tempting to relate this result to the simple fact that the particles with negative spin are more prone to go out of the absorbing boundary at $x = 0$. However, quite remarkably, $Q^\infty(t)$ vanishes in the absence of thermal diffusion, implying that there are equal numbers of positive and negative spin particles in the system. As shown in the Figure 9, the magnetisation is positive at larger $x$ values; as we reduce $x$ beyond some point it becomes negative and for $D = 0$ monotonically decreases to its minimum at $x = 0$, signifying a discontinuity at the absorbing end. Further from Eqs. (66)-(67), $Q_{\mathrm{tr}}(0,t) = -P_{\mathrm{tr}}(0,t)$ for $D = 0$, which implies that only the negative spin particles cross the origin. Consequently $P^+(0,t) = 0$ [48, 49], and discontinuity arises in $P$ (more specifically $P^-$). The above corroborates the general prevalence of negative spin particles near the origin before these are eventually absorbed. Yet, the net balance of positive and negative spin particles is maintained in the system.

The picture is considerably revised in the presence of even a small thermal diffusion. For very small $x$ the time to hit the absorbing boundary is small. But at such small times diffusion dominates over drift, and therefore it compels a larger fraction of the particles near $x = 0$ to get absorbed. However the region near $x = 0$ is majorly populated by the negative spin particles, implying their enhanced elimination from the system because of the diffusion induced absorption consistent with the flux reduction for $s_+ = 0$ ($m_0 = -1$) discussed after Eq. (70). This leads to an imbalance of the two species of particles. This also results in removing the discontinuity at the boundary through the formation of the kinetic boundary layer, which in this case is a depletion layer, of width $k_a^{-1} \sim D/v$ for small $D$. As a whole, diffusion has a complex and often nonnegligible effect on the behaviour of active particle dynamics. Notably, the bulk density

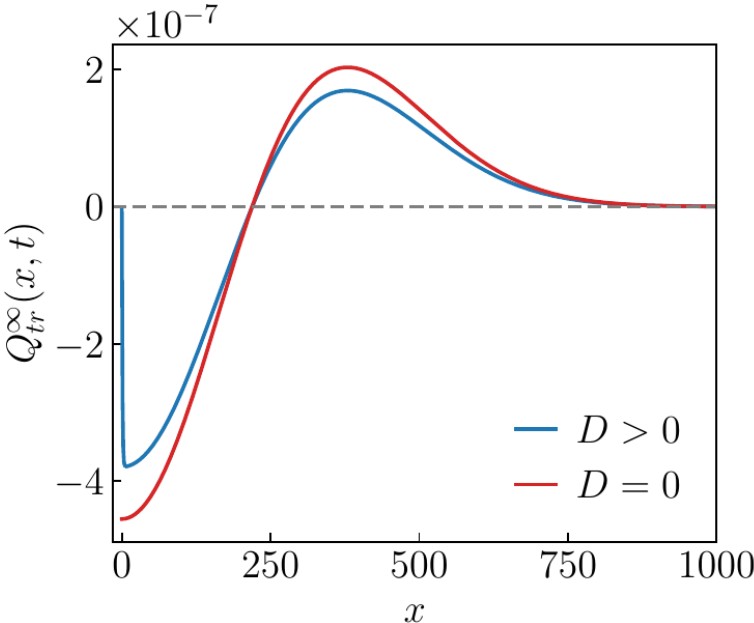

Figure 9: Comparison of magnetisation density using Eq. (67) for $D > 0$ and $D = 0$ cases. Here $v = 1$ and $x_0 = 1$. We have taken $\{D = 1, \omega = 0.1\}$, and $\{D = 0 \, \omega = 1/12\}$ such that both cases are activity dominated and the effective diffusivities are also the same ($D_e = 6$).

profile in Eq. (66) deviates from the absorbing condition at $x = 0$ by an amount proportional to $\frac{v^2}{\omega \sqrt{2\omega D_e}}$, i.e. the Milne length, which is precisely compensated by the boundary layer.

## 5 Universality in the large time distribution

The density profile for the absorbing boundary problem of underdamped passive dynamics resembles closely the distribution in Eq. (66), except that in the former case the boundary layer structure is richer and a boundary discontinuity is present. The solution for the underdamped case is complicated and originally given in the Laplace space [59]. However it is relatively simpler to extract the approximate large time behaviour, which is shown in the Appendix C. Note that the underdamped passive problem is exactly equivalent to the absorbing boundary problem of active Ornstein-Uhlenbeck process (AOUP) in the absence of translational noise and external forces, and so are the solutions. The striking resemblance of such disparate processes, one of which is a passive process, with the only common point being the presence of an additive exponentially correlated noise evokes a general form for the large time distribution in presence of an absorbing barrier,

$$
\begin{aligned}
P_{tr}^\infty(x, t; x_0) &= \frac{1}{\sqrt{4\pi D_e t}} \left\{ e^{-\frac{(x-x_0)^2}{4D_e t}} - e^{-\frac{(x+x_0)^2}{4D_e t}} \right\} \\
&+ \frac{l_M}{2\sqrt{\pi}(D_e t)^{3/2}} (x + x_0) \, e^{-\frac{(x+x_0)^2}{4D_e t}} \\
&+ \frac{u_0 \tau_p}{4\sqrt{\pi}(D_e t)^{3/2}} \left\{ (x - x_0) \, e^{-\frac{(x-x_0)^2}{4D_e t}} + (x + x_0) \, e^{-\frac{(x+x_0)^2}{4D_e t}} \right\} \\
&+ \text{Boundary layer,}
\end{aligned}
\tag{72}
$$

where $u_0$ is the average initial velocity, $\tau_p$ is the noise correlation time or persistence time, and $l_M$ is the Milne length. Clearly, the distribution has three components, $P_{tr}^\infty(x, t) = P_{BM} + P_{CS} + P_{BL}$, where $P_{BM}$ is the passive-like contribution with an effective diffusivity, $P_{BL}$ is the dynamics

dependent boundary layer, and $P_{\rm CS}$ is proposed as a *universal scaling form* emergent due to the coloured noise, $P_{\rm CS}(x,t;x_0) = \frac{l_M}{D_e t}\,\psi(\frac{x}{\sqrt{D_e t}}; \frac{x_0}{\sqrt{D_e t}})$ with,

$$\psi(y;y_0) = \frac{u_0\,\tau_p/l_M}{4\sqrt{\pi}}\left\{(y-y_0)\,e^{-\frac{(y-y_0)^2}{4}} + (y+y_0)\,e^{-\frac{(y+y_0)^2}{4}}\right\} + \frac{y+y_0}{2\sqrt{\pi}}\,e^{-\frac{(y+y_0)^2}{4}}. \qquad (73)$$

The scaling function, however, depends on a parameter $\frac{u_0\,\tau_p}{l_M}$ that takes different values for different initial conditions and nature of the noise. For RTP, $u_0 = m_0 v$, $\tau_p = (2\omega)^{-1}$, and $l_M = \frac{v}{\sqrt{2\omega D_e}}\,l_p$ where $l_p = \frac{v}{2\omega}$ is the persistent length scale. For the nonthermal ($D = 0$) RTP, $l_M = l_p$, the boundary layer vanishes and a boundary discontinuity is created. $l_M$ changes nontrivially in the presence of diffusion which is related to the strength of the boundary layer (Eq. (66)). For the underdamped passive motion, $D_e = D$, $\tau_p = \frac{m}{\gamma}$ and $l_M \approx 1.46\,l_p$ with $l_p = \sqrt{D\,\tau_p}$ , where $m$ is the mass, $\gamma$ is the damping coefficient, and $D$ is the thermal diffusivity. For initial conditions such that $u_0 = 0$, the scaling function does become universal.

A nonzero Milne length implies that the particle far from the boundary sees the absorbing condition at $x = -l_M$ instead of $x = 0$. However in Eq. (66) or Eq. (72), the quantity $P_{\rm tr}^\infty - P_{\rm BL}$ doesn't exactly vanish at $-l_M$ and is off from zero by a tiny amount $\sim l_M^2 t^{-3/2}$ or $l_M u_0\tau_p t^{-3/2}$. For RTP such terms occur from $O(L^{-2})$ corrections in the large-$L$ expansion of the spectrum and are omitted. Motivated from Eq. (C.2) for the underdamped passive case, we incorporate these and combine with $P_{\rm CS}$ and $P_{\rm BM}$ to write the distribution at large times $\sqrt{D_e t} \gg u_0\tau_p, l_M$ in a particularly simple form,

$$P_{\rm tr}^\infty(x,t;x_0) - P_{\rm BL} \approx \frac{1}{\sqrt{4\pi D_e t}}\left[e^{-\frac{(|x-x_0|\pm u_0\tau_p)^2}{4D_e t}} - e^{-\frac{(x+x_0+2l_M+u_0\tau_p)^2}{4D_e t}}\right], \qquad (74)$$

where $\pm$ is for $x \lessgtr x_0$ respectively. This particular form correctly takes into account the absorbing condition at $x = -l_M$, and is proposed as the *universal distribution* for the absorbing boundary problem for dynamics with colored noise with short-ranged correlations. Using simulations we have checked the validity of Eqs. (74) for nonthermal ABP as well, which is shown in the Appendix D.

## 6 Conclusion

Analytical results discussed in this paper for boundary driven noninteracting RTPs provide several new insights for nonequilibrium processes subject to an exponentially correlated noise. For zero boundary magnetisations, which is closest to the passive case, the steady state current satisfy the Fourier law, and in the large $L$ limit both the bulk density profile and current goes over to the usual diffusive behaviour. Notably, in contrast to the passive case, a kinetic boundary layer and finite bulk magnetisation emerges in the system, and the bulk density acquires important corrections equivalent to the substitutions: $x \to x + l_M$ and $L \to L + 2\,l_M$, reminiscent of an effective absorbing conditions outside the boundaries. The scenario with nonzero boundary magnetisation is however starkly different in almost all aspects: The Fourier law is violated, the bulk density and the Milne length can no longer be related to an effective boundary condition, and thermal diffusion acquires a profound role. While nontrivial boundary features are generic to active processes, the complex interplay of the active motion and thermal noise leads to unexpected current reversal in the presence of diffusion; surprisingly, the current also shows a nonmonotonous behaviour as the diffusivity and activity parameters are tuned. Whether such current reversal and nonmonotonous transport are generic to boundary driven active systems will be worth studying.

In presence of finite boundary magnetisations a Seebeck-like effect is identified. In the equilibrium Seebeck effect, the particles carry charge and also energy, the latter being distributed

according to the temperatures at the boundaries. It is plausible that if the particles carry more than one attributes relevant to the dynamics, as in the underdamped as well as the active cases, such effects might occur. An intriguing question is whether magnetisation, which doesn't have any passive counterpart nor has an associated conserved flux, may still be regarded as a thermodynamic force.

In regard to the dynamics, the boundary value problem reduces to a problem with absorbing boundaries at both ends. This enables us to find the spectrum using a reflection symmetry. For large $L$, the eigenvalues and eigenfunctions of the time evolution operator have been found analytically and the full initial value problem is solved at large times $t \gg \omega^{-1}$. The spectrum is arranged in two distinct bands, one around $\lambda = 0$ which determines the long time behaviour, and the other around $\lambda = -2\omega$. The relaxation rate shows an interesting crossover, being a constant for small systems and passive-like in the thermodynamic limit. It is argued that the crossover is related to the crossing of the bands[51]; a quantitative study of the band structure for finite $L$ and analytical determination of the crossover point $L_c$ would be of interest. At large but intermediate times ($\omega^{-1} \ll t \ll L^2/D_e$) the dynamical behaviour mimics the process on a semi-infinite line. In this case the bulk distribution carries strong and often dominant *intrinsically active* signatures not captured by an effective passive description. Many of the features are present in several other systems e.g. the underdamped passive dynamics, the AOUP, and the ABP, where the main quantitative difference occurs in the detailed structure of the boundary layer and the Milne length.

The striking similarity of the distribution in the absorbing boundary problem of such diverse systems points to a new universality in the physics induced by exponentially correlated noise. The unconstrained motions of these particles exhibit diffusive behaviour at large times; similar conclusion holds for reflecting boundaries [28]. The active and passive processes are however very different in the presence of interaction, and a set of model independent features for active single-file dynamics were found recently [60]. Contrary to both of the above, in the absorbing boundary problem the proposed universal distribution in Eq. (74) is new and different from the diffusive one, and it appears to hold for any exponentially correlated noise *including* the passive underdamped motion. We further expect that the steady state of the boundary driven problem in these cases will have a structure similar to Eq. (7), i.e. sum of a linear profile and boundary layers [56, 57]. We must however be cautious that such similarities of active and underdamped passive cases might be lost in presence of trapping and other forces, other noises, or interactions [4]. This is because the active motion considered here is overdamped which is different from the underdamped motion in general, and it would be worthwhile to look into the interplay of inertia and active noise.

Keeping the differences in mind, it seems plausible that the formation of a boundary layer in presence of fluxes at boundaries is generic for motions with coloured noise. For that matter the noise having continuous values might be important, specifically for the athermal case. The one dimensional athermal AOUP gives rise to boundary layers (along with a boundary discontinuity), while the athermal RTP results in a boundary discontinuity only which is smoothened to a boundary layer in presence of diffusion. For the RTP in a 1D box with reflecting boundaries a delta peak occurs at the walls that is converted to an exponential boundary layer for $D > 0$ [13]. On the other hand, in 2-dimension where the orientation of RTP takes continuous values, a boundary layer is reported for reflecting boundaries even when thermal noise is absent [62], and we would expect a kinetic boundary layer for absorbing boundary. Studying absorbing boundary problems with other realisations of continuous valued coloured noise would be useful in this regard.

---

[4]For example the steady state of a trapped athermal AOUP is different from that of a trapped underdamped passive particle [61].

Apart from the boundary layer which is dynamics dependent, it is instructive to investigate the extent of validity of the proposed universality of the late time density profile in Eqs. (73)-(74), e.g. in the underdamped active processes. Further, which of the emergent features are affected in presence of a translational noise is a relevant question. We speculate that the structure of the density profile will remain the same; however the additional length scales introduced because of the thermal noise would nontrivially change the Milne length and will also alter the boundary layer and particle demography so as to remove the boundary discontinuity present in the nonthermal cases. A detailed and more general understanding of the distribution in Eqs. (73)-(74) will be important in gaining insights into the nature of such processes. The 'fine-tuning' of the boundary layer to the Milne length in presence of diffusion, as observed both in the steady state and the late time dynamics, points to a common origin of these nonuniversal features. This curious question is left for future studies.

In conclusion, boundary driven active dynamics provides an elementary physical mechanism for diverse nonequilibrium phenomena − some of which are reported here for the first time, while the others were found previously in disparate settings. This work specifically elucidates the interplay of thermal noise with activity, highlights new features in boundary driven transport, and suggests a novel universality for the motion involving coloured noise. Whether these features survive when the noise correlation is non-exponential yet short ranged, as well as the qualitative changes introduced by interactions, will be of particular interest.

## Acknowledgements

AD acknowledges discussions with Soumyadip Banerjee.

## A    Exit probability of RTP at the left boundary

The exit probabilities $E_{\pm}(x)$ for an RTP is obtained by solving two coupled backward equations with boundary conditions, $E_{\pm}(0) = 1, E_{\pm}(L) = 0$ (see Eq. (51) of [13] with signs interchanged). The equations are identical to that satisfied by $P_{\mp}(s)$ in the steady state and for zero boundary magnetisations the density and magnetisation profiles can be written as,

$$P(x) = P_1 - \Delta P \left[ E_+(x) + E_-(x) \right], \ Q(x) = \Delta P \left[ E_+(x) - E_-(x) \right], \tag{A.1}$$

where $\Delta P = P_1 - P_0$. This relation is used for simulating the steady state distributions. It however turns out that, as the tumble dynamics mixes the spin states, the correspondence between the exit probabilities and the steady state distributions are lost for general boundary conditions.

Note that the results for $E_{\pm}$ reported in Eqs. (C5) of [13] carry some typo. The correct result is,

$$E_m(x) = \frac{(1 + e^{-\mu L})\,x - \frac{v}{2\omega}(m - \frac{v}{\mu D})\left[1 - e^{-\mu x}\right] + \frac{v}{2\omega}(m + \frac{v}{\mu D})\left[e^{-\mu(L-x)} - e^{-\mu L}\right]}{L\left(1 + e^{-\mu L}\right) + \frac{v^2}{\omega \mu D}\left(1 - e^{-\mu L}\right)}. \tag{A.2}$$

Using $m = \pm 1$ one can find the expressions for $E_{\pm}$.

# B   Expression for the left eigenvectors of $\mathcal{L}$ in the $\lambda = 0$ band and the distribution

We find the left eigenvectors $\langle \psi_n |$ of $\mathcal{L}$ by noting that it is just the transpose of the right eigenvector of $\mathcal{L}^\dagger \equiv \mathcal{L}|_{\partial_x \to -\partial_x}$:

$$
\langle \psi_n(x)| \;=\; |\phi_n(x,-v)\rangle^T \approx \alpha'_n \Bigg( 2\,i\,\sin\frac{n\pi x}{L} \begin{bmatrix} 1 \\ 1 \end{bmatrix}^T + i\,\frac{n\pi v}{\omega L}\cos\frac{n\pi x}{L}\begin{bmatrix} 1 \\ -1 \end{bmatrix}^T
$$

$$
- \frac{n\pi v}{\omega L}\Bigg\{ (1 - \frac{v}{\sqrt{2\omega D_e}})\sin\frac{n\pi x}{L} - i\frac{v}{\sqrt{2\omega D_e}}(1-\frac{2x}{L})\cos\frac{n\pi x}{L}\Bigg\}\begin{bmatrix} 1 \\ 1 \end{bmatrix}^T
$$

$$
- i\frac{n\pi v}{\omega L}\,(1+\frac{v}{\sqrt{2\omega D_e}})\Bigg\{ (-1)^{n-1}e^{-k_a(L-x)}\begin{bmatrix}\tilde{a}_n^{(1)} \\ 1\end{bmatrix}^T + e^{-k_a x}\begin{bmatrix} 1 \\ \tilde{a}_n^{(1)}\end{bmatrix}^T\Bigg\}\Bigg). \tag{B.1}
$$

$$
\Rightarrow \int_0^L dx\,\langle\psi_n(x)|\phi_m(x)\rangle = -4\,\alpha'_n\alpha_n L\left(1 - i(i+n\pi)\frac{2\,l_M}{L}\right)\delta_{mn}\;,\; l_M = \frac{v^2}{2\omega\sqrt{2\omega D_e}} \tag{B.2}
$$

up to first sub-leading order in $L$. Invoking the orthonormality condition of the eigenfunctions, we obtain,

$$
\alpha'_n\alpha_n \approx -\frac{1}{4L}\left(1 + i(i+n\pi)\frac{2\,l_M}{L}\right). \tag{B.3}
$$

For the initial condition, $|P(x,0)\rangle = \delta(x-x_0)\begin{bmatrix} s_+ \\ s_-\end{bmatrix}$, we find,

$$
c_n = \int_0^L dx\,\langle\psi_n(x)|P(x,0)\rangle \approx \alpha'_n\Bigg[ 2i\sin\frac{n\pi x_0}{L} + \frac{in\pi v}{\omega L}(s_+ - s_-)\cos\frac{n\pi x_0}{L}
$$

$$
- \frac{n\pi v}{\omega L}(1 - \frac{v}{\sqrt{2\omega D_e}})\sin\frac{n\pi x_0}{L} + \frac{in\pi v^2}{\omega L\sqrt{2\omega D_e}}(1 - \frac{2x_0}{L})\cos\frac{n\pi x_0}{L} \tag{B.4}
$$

$$
- \frac{in\pi v}{\omega L}(1 + \frac{v}{\sqrt{2\omega D_e}})\Bigg\{ (-1)^{n-1}e^{-k_a(L-x_0)}(\tilde{a}_n^{(1)}s_+ + s_-) + e^{-k_a x_0}(s_+ + \tilde{a}_n^{(1)}s_-)\Bigg\}\Bigg].
$$

Using Eqs. (B.3) and (B.4) in Eq. (62), we obtain Eq. (64) for the propagator to $O(L^{-1})$. To find a closed-form expression for times $\omega^{-1} \ll t \ll L^2/D_e$, we define the following quantities corresponding to the summations that occur in Eq. (64) and use the Poisson summation formula:

$$
A(x_1,x_2) \equiv \sum_{n\geq 1} e^{-D_e t\frac{n^2\pi^2}{L^2}}\sin\frac{n\pi x_1}{L}\sin\frac{n\pi x_2}{L} \approx \frac{L}{2\sqrt{4\pi D_e t}}\left[e^{-\frac{(x_1-x_2)^2}{4D_e t}} - e^{-\frac{(x_1+x_2)^2}{4D_e t}} - e^{-\frac{(x_1+x_2-2L)^2}{4D_e t}}\right]
$$

$$
= L\,a_L(x_1,x_2),
$$

$$
B(x_1,x_2) \equiv \sum_{n\geq 1} e^{-D_e t\frac{n^2\pi^2}{L^2}}\frac{n\pi}{L}\sin\frac{n\pi x_1}{L}\cos\frac{n\pi x_2}{L}
$$

$$
\approx \frac{\pi L}{(4\pi D_e t)^{3/2}}\left[(x_1+x_2)e^{-\frac{(x_1+x_2)^2}{4D_e t}} + (x_1-x_2)e^{-\frac{(x_1-x_2)^2}{4D_e t}} + (x_1+x_2-2L)e^{-\frac{(x_1+x_2-2L)^2}{4D_e t}}\right]
$$

$$
= L\,b_L(x_1,x_2)
$$

$$
C(x_1) \equiv \sum_{n\geq 1} e^{-D_e t\frac{n^2\pi^2}{L^2}}\frac{n\pi}{L}\sin\frac{n\pi x_1}{L} \approx \frac{\pi L}{2(4\pi D_e t)^{3/2}}\left[x_1 e^{-\frac{x_1^2}{4D_e t}} + (x_1-2L)e^{-\frac{(x_1-2L)^2}{4D_e t}}\right]
$$

$$
= L\,c_L(x_1) \tag{B.5}
$$

The terms in Eq. (64) that involve $(-1)^n$ are just $C(L - x_1)$, since,

$$C(L - x_1) = -\sum_{n \geq 1} (-1)^n e^{-D_e t \frac{n^2 \pi^2}{L^2}} \frac{n\pi}{L} \sin \frac{n\pi x_1}{L}.$$

The $L$-dependent terms in the square brackets, which are very small for large $L$, are kept in view of the symmetries satisfied by the discrete summation. These also preserve the symmetries of the density and magnetisation profile:

$$\begin{aligned} P(x, t; x_0, m_0) &= P(L - x, t; L - x_0, -m_0), \\ Q(x, t; x_0, m_0) &= -Q(L - x, t; L - x_0, -m_0). \end{aligned} \tag{B.6}$$

Using Eq. (B.5) we rewrite Eq. (64) in terms of $a_L$, $b_L$ and $c_L$,

$$\begin{aligned} |P_{\mathrm{tr}}(x, t)\rangle \;\approx\; & \frac{1}{2} \Bigg( 2\, a_L(x_0, x) \begin{bmatrix} 1 \\ 1 \end{bmatrix} + m_0 \frac{v}{\omega} b_L(x, x_0) \begin{bmatrix} 1 \\ 1 \end{bmatrix} - \frac{v}{\omega} b_L(x_0, x) \begin{bmatrix} 1 \\ -1 \end{bmatrix} \\ & + \frac{v^2}{\omega \sqrt{2\omega D_e}} \left\{ \left(1 - \frac{2x_0}{L}\right) b_L(x, x_0) + \left(1 - \frac{2x}{L}\right) b_L(x_0, x) \right\} \begin{bmatrix} 1 \\ 1 \end{bmatrix} \\ & - \frac{v}{\omega} \left\{ \left(\frac{v}{\sqrt{2\omega D_e}} + m_0\right) e^{-k_a x_0} c_L(x) + \left(\frac{v}{\sqrt{2\omega D_e}} - m_0\right) e^{-k_a(L - x_0)} c_L(L - x) \right\} \begin{bmatrix} 1 \\ 1 \end{bmatrix} \\ & + \frac{v}{\omega} \left(1 - \frac{v}{\sqrt{2\omega D_e}}\right) \left\{ e^{-k_a x} c_L(x_0) \begin{bmatrix} 1 \\ a^{(1)} \end{bmatrix} + e^{-k_a(L - x)} c_L(L - x_0) \begin{bmatrix} a^{(1)} \\ 1 \end{bmatrix} \right\} \Bigg), \quad \text{(B.7)} \end{aligned}$$

which gives an approximate closed form expression of the propagator in the large $L$ limit in terms of known functions. We obtain Eq. (65) in the limit $L \to \infty$. It is evident that unlike in each of the eigenstates, the 'active' contributions appear at the leading order in the general time-dependent distribution.

## C    Large time distribution of a passive underdamped Brownian motion in presence of an absorbing barrier

The absorbing boundary problem of a one dimensional underdamped Brownian motion is surprisingly challenging. The problem was posed in 1945 [63] and finally solved after four decades [59, 64]. Here we are interested in the late time behaviour of the distribution of position of the unbiased underdamped motion. The full solution is quite complex and is originally given in the Laplace space (Eq. (4.4) of [59]):

$$\begin{aligned} \widetilde{P}_{x_0, v_0}(x, v; s) \;=\; & \frac{e^{-v^2/2}}{\sqrt{8\pi}} \sum_{n=0}^{\infty} \frac{e^{-q_n|x - x_0|}}{q_n} f_n^{\mp}(v) f_n^{\mp}(v_0) \\ & - \frac{e^{-v^2/2}}{\sqrt{32\pi}} \sum_{m,n=0}^{\infty} \frac{\sigma_{mn}}{q_m q_n} e^{-q_m x - q_n x_0} f_m^+(v) f_n^-(v_0), \quad \mp \text{ is for } x \lessgtr x_0, \quad \text{(C.1)} \end{aligned}$$

where $q_n = \sqrt{n + s}$, $\sigma_{mn} = \frac{1}{q_m + q_n} \frac{1}{Q_m Q_n} = \sigma_{nm}$ with $Q_n = \lim_{N \to \infty} \frac{\sqrt{n! N!} \exp(2q_n \sqrt{N + 1})}{\prod_{r=0}^{N+n}(q_r + q_n)}$, and $f_n^+(v) = f_n(v) = \frac{e^{v^2/4}}{\sqrt{n!}} \left[ (-1)^n e^{z^2/4} \frac{d^n}{dz^n} e^{-z^2/2} \right]_{z = 2q_n - v}$, $f_n^-(v) = f_n(-v)$. Here all the variables are non-dimensionalised, $t \to t\, \gamma/m$, $x \to x \sqrt{\gamma/(mD)}$, $v \to v \sqrt{m/(D\gamma)}$, $m$ is the mass of the particle. To find the distribution at large times we need to extract the leading behaviour of $\widetilde{P}$ in the $s \to 0$ limit. In this limit, the summations with only $m = 0$ or $n = 0$ will contribute in Eq. (C.1), and the different quantities are evaluated as,

I. $f_0(v) \approx e^{-s+v\sqrt{s}}$, $f_n(v) \approx \frac{\Phi_n(v)}{\sqrt{n!}} e^{-n+v\sqrt{n}}$ for $n > 0$, where $\Phi_n(v)$ is a polynomial of degree $n$.

II. The $Q_n$'s are evaluated using Euler-Maclaurin formula: $\sum_{r=1}^{N} f(r) = \int_{1}^{N} f(x)dx + \frac{f(1)+f(N)}{2} + \mathcal{R}_N$, where $\mathcal{R}_N$ is the residue. Without considering $\mathcal{R}_N$ we find,

$$Q_0 \approx \frac{1}{2\sqrt{s}} e^{\frac{3}{2}\sqrt{s}}, \quad Q_n(s) \approx \Big[\frac{n!}{n(1+\sqrt{n})}\Big]^{\frac{1}{2}} e^{\frac{n}{2}+\sqrt{n}} e^{-\sqrt{s/n}}$$

for $n > 0$. If we take the residue term into account, the exponent $\frac{3}{2}$ in $Q_0(s)$ would be slightly modified; e.g. the first correction would give an exponent $\frac{3}{2} - \frac{1}{24} \approx 1.458$. In $Q_n$, the residue would introduce a slowly varying prefactor $b_n : Q_n \to b_n Q_n$.

Putting these together, and integrating out the velocity, we find the distribution of the particle's position in the Laplace space,

$$\widetilde{P}_{x_0,v_0}(x; s) \approx \frac{e^{-\frac{3}{2}s}}{2\sqrt{s}} \Big[e^{-(|x-x_0|\pm v_0)\sqrt{s}} - e^{-(x+x_0+2l_M+v_0)\sqrt{s}}\Big] + \widetilde{P}_{BL}(x; s), \qquad (C.2)$$

where $l_M \approx 1.458$, $\pm$ is for $x \lessgtr x_0$, and the boundary layer profile is given by,

$$\begin{aligned}
\widetilde{P}_{BL}(x; s) &\approx \frac{1}{2}\sum_{n=0}^{\infty}\Big[\frac{1+\sqrt{n}}{n}\Big]^{\frac{1}{2}} \frac{e^{-\frac{3}{2}n-\sqrt{n}}}{n!\, b_n} e^{-(v_0-\frac{1}{\sqrt{n}})\sqrt{s}} \\
&\quad \times \Big[H(n)\, e^{-x\sqrt{n}}\, e^{-(x_0+v_0)\sqrt{s}} + \Phi_n(-v_0)\, e^{-(x_0-v_0)\sqrt{n}}\, e^{-x\sqrt{s}}\Big], \qquad (C.3)
\end{aligned}$$

with $H(n) = \frac{1}{\sqrt{2\pi}}\int_{-\infty}^{\infty} dv\, \Phi_n(v)\, e^{-\frac{v^2}{2}+v\sqrt{n}}$. Plugging in the units and taking the inverse transform, we find the leading time dependence of the density profile for the problem, which is given by Eq. (72).

The distribution is markedly different from that obtained in the usual overdamped problem. It contains a rich 'kinetic boundary layer' structure of a finite skin depth $\sim l_p = \sqrt{mD/\gamma}$ near the absorbing walls. Secondly, far from the wall at large times the distribution takes a scaling-like form $P_{sc} = P_{x_0,v_0}^{\infty}(x,t) - P_{BL}$ which corresponds to a spatial shift $x \to x + l_M$. $P_{sc}$ satisfies the absorbing condition at $x = -l_M$ instead of $x = 0$ [54], i.e. $P_{sc}(-l_M, t) = 0$, implying that $l_M$ is the Milne length. In [64] the Milne length is exactly determined, $l_M = -\zeta(\frac{1}{2})\, l_p \approx 1.460\, l_p$, a value approximated remarkably well by the Euler-Maclaurin formula. Also note the factor $e^{-\frac{3}{2}s}$ in Eq. (C.2) that corresponds to a temporal shift, $t \to t - \frac{3}{2}\tau_p$ with $\tau_p = m/\gamma$, suggesting that $P_{sc}$ pertains to an initial condition at $t = \frac{3}{2}\tau_p$ instead of $t = 0$ [54].

# D  Distribution of ABPs on a semi-infinite line with absorbing condition at $x = 0$

The equation of motion of an overdamped ABP is given by,

$$\dot{x} = v\cos\theta(t) + \eta(t), \quad \dot{\theta} = \sqrt{2D_r}\,\zeta(t), \qquad (D.1)$$

where $v$ is the self-propulsion speed, $\theta$ is the instantaneous orientation of the particle which takes on continuous values, and $\eta$ is the thermal noise; here $\theta$ evolves as a Brownian particle with (rotational) diffusivity $D_r$. Since $\langle\cos(\theta(t))\cos(\theta(t'))\rangle = e^{-D_r|t-t'|}$, the dynamics of ABP is driven by an exponentially correlated noise.

We simulated the Eq. (D.1) with $\eta(t) = 0$ on a semi-infinite line subject to the absorbing boundary condition at $x = 0$. We have taken random initial orientation such that $u_0 = 0$.

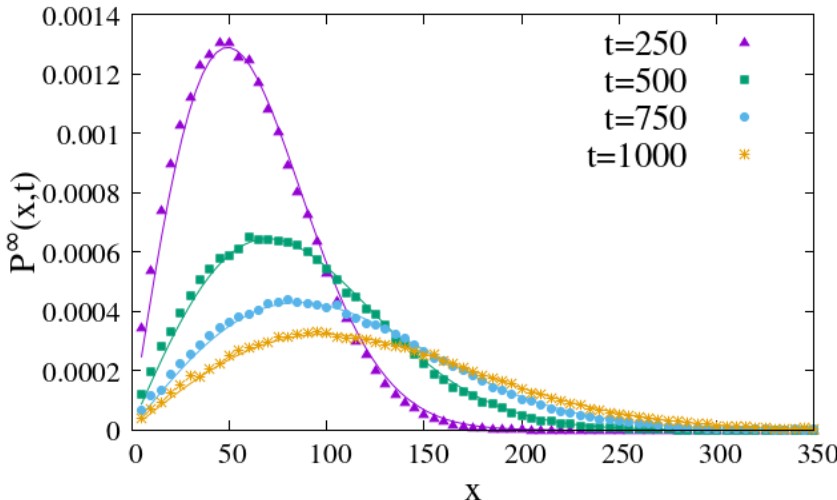

Figure 10: Distribution of athermal ABPs on a semi-infinite line with an absorbing barrier at $x = 0$. The data is shown with particle parameters $v = \sqrt{2}, D_r = 0.2$ and initial condition $x_0 = 1.0, u_0 = \langle \cos(\theta(0)) \rangle \approx 0$. The line corresponds to the expression in Eq. (72) except the boundary layer contribution.

The distribution at large times is captured remarkably well with $P^\infty(x,t) - P_{\mathrm{BL}}$ given in Eqs. (72)-(73), with $D_e = \frac{v^2}{2 D_r}, l_M \approx 0.8 \frac{v}{D_r}$. The result is shown in Figure 10. This corroborates the proposed universality of the large time distribution. Note that $P(x,t)$ has a discontinuity at $x = 0$. Since the noise is continuous-valued, we expect a boundary layer to be present that will alter the expression of the distribution near $x = 0$. This is however not taken up in this paper.

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
