# Peer review of "Boundary layers, transport and universal distribution in boundary driven active systems"

_SciPost Physics_

## Round 2 · Referee Report · Anonymous (Referee 1) · 2025-4-16

Strengths

1 - Introducing the concept of active motion in channels connected to active particle reservoir as a playground to explore novel features of nonequilbrium transport
2 - A number of novel analytical results, both for steady state statistics and relaxational dynamics
3 - Interesting physical insight and identification of mechanisms/features that may be feasibly generalised to a broader class of problems

Weaknesses

1 - Notation could be more transparent (see requested changes)
2 - Algebra-heavy sections in main text can be hard to navigate on a first read
3 - A microscopic definition of boundary interactions is not provided, preventing numerical verification of some of the results and potentially obscuring the applicability of the derived results

Report

This work presents a systematic and thorough exploration of (non-interacting) 1D run-and-tumble motion in the presence of boundary driving by “active” reservoirs, which are mathematically defined to impose fixed density and polarity boundary conditions. A number of novel results are derived, including (1) the steady state distribution; (2) the eigenspectrum and time-dependent distribution, with explicit results mostly being obtained in the large system size and long relaxation time limit; (3) a proposed universal form of the large time distribution on the semi-infinite line (with an absorbing boundary) for “macroscopically-diffusive” processes with an exponentially correlated noise, which is shown to hold both for equilibrium (inertial) motion and non-equilibrium self-propelled motion. Interesting parallels are drawn with traditional transport theory, including the definition of an effective Milne length, the violation of Fick’s law in the presence of non-zero boundary polarity and a Seebeck-like effect. The impact of bare diffusion on the establishment of boundary layers in both the steady state and transient distributions is also thoroughly analysed. Indeed, the authors put an admirable amount of effort into extracting as much of the relevant physics from their analytical results as possible (with the downside of the text being at times quite verbose). I believe that this work meets the criteria for publication in SciPost Physics, insofar as it will certainly stimulate further research into the non-trivial features of boundary-driven transport in active matter systems. I thus recommend publication after minor adjustments, as detailed below.

Requested changes

1 - Define meaning of a Milne length in physical terms in the context of this work when first introduced
2 - A microscopic mechanism to enforce the boundary conditions discussed in this work would be a valuable addition and allow for future numerical validation/exploration. I wonder if this could be done in a similar vein to [Roberts & Pruessner, PRR 4(3), 033234 (2022)].
3 - Below Eq.(9), could the authors say a few words about the physical interpretation of the of the various effective parameters?
4 - In point V (page 6), what is the meaning of $\overline{Q_1 + Q_1}$?
5 - In Sec.4, I find the distinction between $a_n$ and $\alpha_n$ visually confusing, perhaps a different symbol for either of the two would help? Also in Eq.(20), it should be clarified why $\alpha_n$ is not absorbed into the definition of $| A_n \rangle$.
6 - Below Eq.(34), the two eigenvalues $\lambda=0$ and $\lambda=-2\omega$ are identified. The authors write that the second "corresponds to the tumble dynamics", by which I guess they mean that it is associated with the relaxation of spatially homogeneous polarity fluctuations (phrasing could be clarified). Could the authors comment more explicitly about why only the bands of eigenvalues around these two are deemed relevant for further analysis?
7 - Below Eq.(67): "the last but one term falls...". Is this a typo?
8 - Sec.5 should perhaps more accurately be titled "A proposed universality..."
9 - Above Eq.(74): "to write the distribution at large times..." should include a reference to the equation being referenced, I assume (72)

Recommendation

Ask for minor revision

  • validity: high
  • significance: high
  • originality: good
  • clarity: high
  • formatting: excellent
  • grammar: excellent

Author:  Arghya Das  on 2025-06-27  [id 5603]

(in reply to Report 1 on 2025-04-16)
Category:
answer to question
correction

Comment: "This work presents a systematic and thorough exploration of (non-interacting) 1D run-and-tumble motion in the presence of boundary driving by ''active'' reservoirs, ... active matter systems. I thus recommend publication after minor adjustments, as detailed below."

Reply: We thank the referee for finding our results novel and recommending the paper for publication. Below is our response.

Comment: "1 - Define meaning of a Milne length in physical terms in the context of this work when first introduced."

Reply: Milne length in the absorbing boundary (or first passage) problem quantifies the difference of the apparent and real positions of the boundary. It is now defined at the Introduction.

Comment: "2 - A microscopic mechanism to enforce the boundary conditions discussed in this work would be a valuable addition and allow for future numerical validation/exploration. I wonder if this could be done in a similar vein to [Roberts & Pruessner, PRR 4(3), 033234 (2022)]."

Reply: Indeed the reservoir-particle interaction at the boundaries, in particular in the presence of fluxes, is an important and challenging issue. We think that it deserves a study in itself and is beyond the scope of the present paper. We have added an appendix (now Appendix A) outlining few possibilities that might be explored to this end. We also thank the referee for suggesting an interesting article, which is now cited.

Comment: "3 - Below Eq.(9), could the authors say a few words about the physical interpretation of the of the various effective parameters?"

Reply: Here $D_e$ is the effective diffusivity in the presence of self-propulsion, $\mu^{-1}$ is the width of boundary layer, $M$ is related to the ratio between Milne length and persistence length in the large system size limit and $B_0$ is an integration constant related to the strength of the boundary layer. These are now added after Eq. (9).

Comment: "4 - In point V (page 6), what is the meaning of $\overline{Q_0+Q_1}$?"

Reply: It is just $(Q_0+Q_1)$. We have now changed the notation.

Comment: "5 - In Sec.4, I find the distinction between $a_n$ and $α_n$ visually confusing, perhaps a different symbol for either of the two would help? Also in Eq.(20), it should be clarified why αn is not absorbed into the definition of $|A_n⟩$."

Reply: We have now repalced $\alpha_n$ by $\beta_n$.
In principle, $\alpha_n$ (now $\beta_n$) can be absorbed in $|A_n⟩$. Note that, $|A_n\rangle$ also satisfies an eigenvalue equation. By keeping them separate, the form of $|A_n\rangle$ remains simpler and it makes the calculation cleaner. Also, their role is a bit different: $\alpha$'s (now $\beta$) depend on the system specifications like boundary conditions and can be regarded as the amplitude of the spatial mode, while $|A_n\rangle$ is instrinsic to the dynamics and is regarded as part of the spatial mode itself, as apparent in Eqs. (26)-(28).

Comment: "6 - Below Eq.(34), the two eigenvalues $λ=0$ and $λ=−2ω$ are identified. The authors write that the second "corresponds to the tumble dynamics", by which I guess they mean that it is associated with the relaxation of spatially homogeneous polarity fluctuations (phrasing could be clarified). Could the authors comment more explicitly about why only the bands of eigenvalues around these two are deemed relevant for further analysis?"

Reply: Since $\omega$ is the tumble rate, $\omega^{-1}$ gives a time scale for the tumble dynamics and $2\omega$ is the corresponding relaxation rate. This is largely related to the relaxation of initial magnetisation (not discuss here). We now rephrased the sentence.
We have noticed that $\lambda = 0$ and $\lambda=-2\omega$ satisfy the equation $det[S]=0$. Small time scale behaviour is governed by the tumble dynamics which corresponds to $\lambda=-2\omega$ and nearby eigenvalues while the long time scale dynamics is governed by the eigenvalues near $\lambda=0$. These two values are intrinsic to the dynamics and we worked on the guess that the eigenvalues for large $L$, infinite in number, are distributed around these. We could not immediately rule out other such central values of $\lambda$ that would satisfy the above determinant equation, but we could neither identify those physically or from the equations.

Comment: "7 - Below Eq.(67): "the last but one term falls...". Is this a typo?"

Reply: We have removed this sentence.

Comment: "8 - Sec.5 should perhaps more accurately be titled "A proposed universality..." "

Reply: We agree with the suggestion and has changed the title of Sec.5 as "Proposed universality in the large time distribution".

Comment: "9 - Above Eq.(74): "to write the distribution at large times..." should include a reference to the equation being referenced, I assume (72)."

Reply: That's correct. We have now referred to this equation (now Eq. (66)) in the sentence.

---

## Round 2 · Referee Report · Goncalo Antunes (Referee 2) · 2025-4-29

Strengths

1- The discussion is very thorough, no steps are skipped in the mathematical handling, and the results are very deeply analysed 2 - The dynamics as well as the steady-state profiles are studied 3- A novel boundary effect was identified when the average particle direction is kept fixed at the domain boundaries. 4 - A discussion of the impact of the paper's results to the broader field is had 5 - Closed analytical solutions are obtained 6- The authors have made an admirable effort to have the reader understand the material and their methods (e.g. Fig. 5).

Weaknesses

1 - The paper is too detailed: the impact of the big results is lost because the reader must wade through many details of less importance
2 - No adimensionalization is performed and no units are mentioned.
3 - The paper makes a number of large-sounding claims that I believe when investigated closer, are not as insightful as they are made to be / are based on very loose analogies. An example is the supposed breaking down of Fourier's law, when in reality there is no temperature gradient or heat flows in the system. What the authors mean is that the flux of particles is not linearly proportional to the density gradient, which is well known in active system.
4 - I am unconvinced by a number of logical steps made during the discussion
5 - The notation is dense, with many quantities being defined, without a discussion of their physical significance

Report

The paper has a clear and interesting research question and explores it thoroughly. The topic and results are of general interest to the active matter / statistical physics community. One notices that the authors have done their due diligence when investigating the consequences of their results, and I find the boundary layers that form due to the boundary conditions to be interesting results.

However, the paper is very dense, and often gets lost in details. I would suggest that a large chunk of the mathematical derivation be placed in the supplementary material, with the main results being mentioned in the main text.

I found some claims / results that strike me as weird: for example Eq. (67) describes the average orientation of the particles as a function of time. However (as noted by the authors), it does not depend on the initial average orientation. This is very strange, as I would expect this initial orientation to be the limit of Eq. (67) as time goes to zero.

As a minor point: a number of typos must be corrected (such as the lower case "j" and "c" in "janus particle" or "casimir effect", or the subscripted $(x,t)$ in the caption of Fig. 1 ).

While I believe expectations criteria #2 (Open a new pathway in an existing or a new research direction, with clear potential for multi-pronged follow-up work) is met, I can only recommend publication after heavy revision / clarification from the authors.

Requested changes

1- A discussion should be had on the real, physical realization of magnetization control. Do the authors know of a system where boundary magnetization can be controlled?

2 - In page 3, the authors mention "diffusive for large L" but have not yet introduced what L is.

3- the Milne length should be introduced: what is its physical meaning?

4- the authors should be careful when referring to particles travelling in opposite direction as "two species" as such a work invokes in the reader ideas of more qualitative differences (such as different values of $v$, for example).

5- Many variables are introduced in page 5, but very little discussion is had on why these quantities are important to define, and why they are defined that way. For example, what is the physical meaning of $\mu$ or $B_0$?

6- the "c" in "casimir effect", the "j" in "janus particle" should be capitalized. Furthermore, there is a typo in the caption of Fig. 1 where $(x,t)$ is in subscript.

7- statements such as "For $L$ large the current and the density gradient..." should be changed to indicate what scale is being used to compare $L$ with. Meaning, what is $L$ large in comparison to?

8- I think the analogy with the Fourier law and the Seeback effect is very stretched, and ends up overinflating the importance of the results. There is no temperature gradient along the system, and there is no heat flow. What the authors mean is that the flux of particles is not linearly proportional to the gradient of particle density. This is Fick's law, which is well-known to not apply to run-and-tumble particles. A similar analogy states that there is a Seebeck effect, but there is no temperature drop and there is no voltage drop. What the authors means is that a state with zero current implies a balance between the diffusive flow and the active flow. Therefore, imposing magnetisation in zero net current must lead to a gradient in density. This is not surprising. The connection to the Seebeck effect involves a number of analogies to be made, and to be honest, I think hurts the paper more than it helps. It ends up leaving the reader feeling as if they were over-promised something.

9 - point II of page 6 is a bit jumbled. I understand what it is saying, but it's written in a very contrived way. What is a "fine-tuned kinetic boundary"? What does it mean to "restore the boundary condition"?

10 - What is $\overline{Q_0 + Q_1}$ in point V of page 6? I find the last sentence of this point to be also hard to understand

11 - Point VI of page 6 states that "the current governs the nature of the bulk profiles and not the boundary conditions". I do not understand this statement as boundary conditions should uniquely determine the solution.

12 - The discussion of page 6 would greatly benefit from references to Figs. 2 and 3. It is not clear what is meant with current reversal.

13 - A suitable adimensionalization must be made, or the authors must state their units. Statements such as "$v=D=1$" (caption of fig. 8) are very jarring.

14 - While the caption states that the results of Fig. 3 are amplified a factor of $10^4$, this must be made clear in the figure itself.

15 - I am unclear on exactly what simulations were performed in section 3.1. Appendix A seems to be entirely analytical, no?

16 - the large $L$ limit of section 3.1 should be done more rigorously, as it neglects some terms of order $e^{-\mu L}$, but not others. Why are only some terms OK to ignore?

17 - It should be clarified that $A_n^+$ and $A_n^-$ are new quantities that have not been previously defined

18 -Why are the eigenvalues expected to be close to zero when $L$ is large ( section 4.1.1)?

19 - Why is it OK to neglect $e^{-k_a L}$ when $L$ is large, but not $e^{-k_b L}$ (page 13)?

20 - Is $| P_{tr}>$ in Eq. 62 a complex number? According to its definition, it should stay real.

21 - I don't understand why the magnetization of Eq. (71) does not tend to $m_0$ in the limit of $t \rightarrow 0$.

22 - The discontinuity of Fig. 9 is a bit disturbing to me. How compatible is it with the boundary conditions? What steps have been made to ensure there is no error (numerical or otherwise)? If you take the limit of $D \rightarrow 0$, do you see a continuous sharpening of the peak or do you always have a jump?

23 - What does "boundary layer" mean in Eq. 72?

24 - What are the definitions of $P_{BM}$ and $P_{BL}$ in page 20?

Recommendation

Ask for minor revision

  • validity: high
  • significance: high
  • originality: high
  • clarity: ok
  • formatting: perfect
  • grammar: perfect

Author:  Arghya Das  on 2025-06-27  [id 5604]

(in reply to Report 2 by Goncalo Antunes on 2025-04-29)
Category:
answer to question
correction
validation or rederivation

Comment: "The paper has a clear and interesting research question and explores it thoroughly. The topic and results are of general interest to the active matter / statistical physics community...
While I believe expectations criteria 2 (Open a new pathway in an existing or a new research direction, with clear potential for multi-pronged follow-up work) is met, I can only recommend publication after heavy revision / clarification from the authors."

Reply: We thank the referee for finding our manuscript interesting and recommending it for publication after due revisions. We have incorporated necessary changes based on the suggestions and comments of the referee. Below is our pointwise reply.

Comment: "1- A discussion should be had on the real, physical realization of magnetization control. Do the authors know of a system where boundary magnetization can be controlled?"

Reply: Active particles generally show nontrivial boundary features and the work goes with the expectation that the boundary magnetisation is typically nonzero, and can have different values in different reservoir-system set ups. Here it is assumed that in a given set up the boundary magnetisations are maintained at certain values in the presence of boundary fluxes and the consequences are discussed. Magnetisation control is not within the scope and purview of the paper.
At the same time we agree that realistic tuning of the magnetisation is important and would be of help in observing the results. We have now added an appendix (appendix A in the revised manuscript) where we speculate very briefly on a few possibilities to achieve it.

Comment: "2 - In page 3, the authors mention "diffusive for large L" but have not yet introduced what L is."

Reply: $L$ is the system size. We have modified the sentence in the text.

Comment: "3- the Milne length should be introduced: what is its physical meaning?"

Reply: It turns out that in the absorbing boundary (or first passage) problem the apparent positions of the boundary, when seen from a distance, might be different from the actual position. Milne length is just the difference of the real and apparent position of the boundaries. It is now defined at the Introduction.

Comment: "4- the authors should be careful when referring to particles travelling in opposite direction as "two species" as such a work invokes in the reader ideas of more qualitative differences (such as different values of v, for example)."

Reply: We have changed the phrase "two species of particles" to "differently oriented particles".

Comment: "5- Many variables are introduced in page 5, but very little discussion is had on why these quantities are important to define, and why they are defined that way. For example, what is the physical meaning of $μ$ or $B_0$?"

Reply: We have added a discussion on the physical meaning of these variables after Eq. (9).

Comment: "6- the "c" in "casimir effect", the "j" in "janus particle" should be capitalized. Furthermore, there is a typo in the caption of Fig. 1 where (x,t) is in subscript."

Reply: Thanks for pointing these out. The typos are now corrected.

Comment: "7- statements such as "For L large the current and the density gradient..." should be changed to indicate what scale is being used to compare L with. Meaning, what is L large in comparison to?"

Reply: $L$, the system size, is compared to the length scales related to the particle dynamics, e.g. the width of the boundary layer $\mu^{-1}$, persistence length $(l_p=v/2\omega)$. We have now clarified this in the text.

Comment: "8- I think the analogy with the Fourier law and the Seeback effect is very stretched, and ends up overinflating the importance of the results. There is no temperature gradient along the system, and there is no heat flow. What the authors mean is that the flux of particles is not linearly proportional to the gradient of particle density. This is Fick's law, which is well-known to not apply to run-and-tumble particles. A similar analogy states that there is a Seebeck effect, but there is no temperature drop and there is no voltage drop. What the authors means is that a state with zero current implies a balance between the diffusive flow and the active flow. Therefore, imposing magnetisation in zero net current must lead to a gradient in density. This is not surprising. The connection to the Seebeck effect involves a number of analogies to be made, and to be honest, I think hurts the paper more than it helps. It ends up leaving the reader feeling as if they were over-promised something."

Reply: We thank the referee for the comment and observations, which is quite helpful. Mentioning 'Fourier Law' in this section was erroneous and it is now corrected. We agree that the violation of Fick's law is well-known in the active literature. Here we found that, at a global level, this law is modified for the current driven by boundary reservoirs whenever the net boundary magnetisation $Q_0+Q_1$ is nonzero.

That there is a modification to Fick's law is not unexpected. However our contention, which we have tried to put up more clearly in the revised manuscript, is to understand the global current induced by a net nonzero boundary magnetisation. To us this is not at all obvious.
Take for example the left boundary ($x=0$) with density $P_0=P_0^+ + P_0^-$ and magnetisation $Q_0=P_0^+ - P_0^-$. This implies, at that boundary, the particle will have a spin $\pm 1$ with probabilities $s_0^{\pm}=P_0^{\pm}/P_0$ respectively; this would also induce a net local velocity $u=v\,(s_0^+ - s_0^-)=v\,Q_0/P_0$ to the particle. One may think of this as preferential `kicks' or manipulating the particle orientation, localised strictly at the boundaries. For ballistic particles without a tumble dynamics ($\omega=0$) the bulk magnetisation and current are just $\sim Q_0$, which is clearly understood. However for particles with finite tumble rate the picture is quite different.
In a large system the preferential orientation at $x=0$ is not expected to affect the properties in the bulk and the global current because of the tumble dynamics, and in the overdamped situation the local push $u$ would dissipate rapidly; the latter can be immediately checked for a Brownian particle pushed at the boundaries. Yet there is current and bulk magnetisation, but now with a magnitude reduced to $\sim Q_0/L$.
$~~~~$ It is notable that at the white noise limit of the RTP dynamics, $\lim_{v\to \infty,\omega \to \infty} \frac{v^2}{2\,\omega}=D_a$ finite, the profiles develop discontinuities at the boundaries, the bulk magnetisation vanishes as $\omega^{-1/2}\,$ while the current persists. This is understandable since in this case $u \to \infty$, and such localised but infinite impulses on an overdamped Brownian particle can sustain a current through the system. When both $v,\omega$ are finite, the cascade of runs and tumbles somehow manages to retain deep in the bulk an imprint (reduced by $L$) of the orientation imbalance at the boundaries, which in turn keeps preferentially driving the particle and thus sustaining a current. This dual role of magnetisation may require more careful look.

Considering the 'Seebeck-like' effect, we emphasize that this is a real transport effect bearing new features.
In the context of heat transport in active chains, a balance of diffusive and active heat currents were reported earler. Here we argue that this is not only generic at more elementary levels but can be leveraged to extract additional transport behaviour. For example, in the spirit of a thermo-couple, we can add another channel end to end with the channel under consideration (call this circuit an `active-couple') while maintaining net nonzero magnetisation at the junctions. This will generate a particle current in the circuit.
The central element is the current induced by boundary magnetisations, which once there naturally leads to such effects.
$~~~~$ An important point is that the phenomenology of this effect is entirely different from Seebeck effect or other near equilibrium current balance phenomena involving Onsager coefficients. In those cases the thermodynamic potentials induce corresponding forces (gradients) in the bulk of the system; the resulting mass currents balance each other at all points while the forces themselves remain nonzero. For the Seebeck effect, the local temperature gradient induces local voltage (or density) gradient, which slowly adds up to give the global potential drop.
On the contrary, in the zero current condition for boundary driven RTPs, both local density gradient and magnetisation separately vanish in the bulk. And the global effect is maintained by (relatively large) forces confined near the boundaries only. To our knowledge this is a new effect, without a thermodynamic analog. We have now added the explanations in the manuscript. Direct analogy to a Seebeck-like effect is removed, but we still refer to the outward resemblance for familiarity.

Comment: "9 - point II of page 6 is a bit jumbled. I understand what it is saying, but it's written in a very contrived way. What is a "fine-tuned kinetic boundary"? What does it mean to "restore the boundary condition"?"

Reply: We mean that if we extrapolate the linear bulk density profile to the boundaries, it does not match with the boundary conditions. This is also the case for $D=0$ where we actually have boundary discontinuities. However for $D\gt 0$ a boundary layer if formed which compensates the deviation. We have now rephrased the relevant paragraph.

Comment: "10 - What is $\overline{Q_0+Q_1}$ in point V of page 6? I find the last sentence of this point to be also hard to understand."

Reply: $\overline{Q_0+Q_1}$ is same as $(Q_0+Q_1)$. We now changed the notation.
In the said sentence we mentioned the conditions of current reversal without changing the boundary conditions. The key point is, the direction of the current is determined by whether $M$, which is a function of the diffusivity and activity parameters alone, is greater or less than $M^*=\frac{\Delta P}{Q_0+Q_1}$. However, $M$ is also constrained to take values within $(0,1)$, and therefore whenever $M^*\lt1$, we can tune $M$ across $M^*$ by appropriately changing the parameters. Now we have clarified the discussion in the revised manuscript.

Comment: "11 - Point VI of page 6 states that "the current governs the nature of the bulk profiles and not the boundary conditions". I do not understand this statement as boundary conditions should uniquely determine the solution."

Reply: It's true that the boundary conditions uniquely determine the solution. We have found that the nature of the profiles depend sensitively on the direction of current, e.g. change in slope and non-monotonicity, which in turn is not necessarily governed by the boundary condition alone. For instance, keeping the boundary conditions fixed we can generate density profiles of qualitatively different nature for different values of the diffusivity or tumble rate. We have now modified the discussions and also replaced the plots in Fig. 2 to clarify the point.

Comment: "12 - The discussion of page 6 would greatly benefit from references to Figs. 2 and 3. It is not clear what is meant with current reversal."

Reply: Now we have added references to Fig.2 and 3 in the discussion. By the term current reversal we mean a change in the direction of the current, which we now mention clearly.

Comment: "13 - A suitable adimensionalization must be made, or the authors must state their units. Statements such as "v=D=1" (caption of fig. 8) are very jarring."

Reply: Thanks for pointing this out. We haven't resort to any specific unit but it might be chosen as required in a specific system. In the caption of Fig. 8 we have changed it to $v=1,~D=1$.

Comment: 14 - While the caption states that the results of Fig. 3 are amplified a factor of 104, this must be made clear in the figure itself."

Reply: We have made it clear now in the figures.

Comment: "15 - I am unclear on exactly what simulations were performed in section 3.1. Appendix A seems to be entirely analytical, no?"

Reply: We simulate the exit probabilities and used Eq. (A.1) to get the staedy state probabilities. We have now explained the procedure in section 3.1.

Comment: "16 - the large L limit of section 3.1 should be done more rigorously, as it neglects some terms of order $e^{−μL}$, but not others. Why are only some terms OK to ignore?"

Reply: We have only ignored terms which are of order $e^{-\mu L}$ for all $x$. We kept terms like $e^{-\mu (L-x)}$ since it is finite near the right boundary.

Comment: "17 - It should be clarified that $A^+_n$ and $A^−_n$ are new quantities that have not been previously defined."

Reply: Actually we do not need to introduce $A^+,A^-$ and these are now removed.

Comment: "18 -Why are the eigenvalues expected to be close to zero when L is large (section 4.1.1)?"

Reply: For large $L$ the relaxation time is also large, and therefore $\lambda_1$ must be close to zero. Further, in this limit the dynamics of the particle is a priori expected to be close to diffusive, and therefore other `low lying' eigenvalues are close to zero as well. That there is a band near $\lambda=0$ is an assumption made on these expectations, which is consistently verified retrospectively. We now elaborated the sentence (now in section 4.2.1) a bit.

Comment: "19 - Why is it OK to neglect $e^{−k_aL}$ when L is large, but not $e^{−k_bL} $ (page 13)?"

Reply: The answer is related to the last question. Eq. (43) shows that, for small $\lambda$, $k_a$ is finite but $k_b\sim \sqrt{\lambda}\to 0$. This allows us to neglect $e^{-k_a L}$ but not $e^{-k_b L}$.

Comment: "20 - Is $|P_{tr} \rangle$ in Eq. 62 a complex number? According to its definition, it should stay real."

Reply: In Eq.(62), $|P_{\rm tr}\rangle$ contains unknown co-efficient $c_n$ which is calculated in Appendix B (now Appendix D). After substituting the expression of $c_n$, $|P_{\rm tr}(x,t)\rangle$ comes out to be real only.

Comment: "21 - I don't understand why the magnetization of Eq. (71) does not tend to $m_0$ in the limit of $t→0$."

Reply: The magnetization $Q^{\infty}(t)$ given in Eq.(71) (Now Eq. (65)) is calculated from $Q_{\rm tr}(x,t)$ (now Eq. (61)), which is valid for time ranges $\omega^{-1}\ll t \ll L^2/D_e$. To find the short time ($t\ll \omega^{-1}$) behaviour we need to incorporate the band near $\lambda=-2\,\omega$, and then only we can take the $t\to 0$ limit.

Comment: "22 - The discontinuity of Fig. 9 is a bit disturbing to me. How compatible is it with the boundary conditions? What steps have been made to ensure there is no error (numerical or otherwise)? If you take the limit of $D→0$, do you see a continuous sharpening of the peak or do you always have a jump?"

Reply: We want to clarify that Fig.9 is obtained just by plotting the expression given in Eq.(67) (now Eq. (61)) for different set of parameters. And indeed, there is no discontinuity for $D\gt 0$ but a continuous sharpening of the boundary layer as $D$ approaches zero. The discontinuity occurs strictly at $D=0$.

Comment: "23 - What does "boundary layer" mean in Eq. 72?"

Reply: For the models under consideration, there are components in the probability distribution that decays exponentially as one moves away from the absorbing end. These terms don't have a universal form and its structure varies across models. We generically mention such terms as `boundary layer'. We have now mentioned this in the text.

Comment: "24 - What are the definitions of $P_{BM}$ and ${P_BL}$ in page 20?"

Reply: $P_{\rm BM}$ is the passive like contribution to $P_{\rm tr}^{\infty}(x,t;x_0)$ with an effective diffusion constant $D_e$.
\begin{equation}
P_{\rm BM}= \frac{1}{\sqrt{4\pi D_e t}} \left\lbrace e^{-\frac{(x-x_0)^2}{4D_e t}} - e^{-\frac{(x+x_0)^2}{4D_e t}} \right\rbrace \nonumber
\end{equation}
And $P_{\rm BL}$ is the contribution coming from the boundary layers. Generally this term contains an exponentially decaying function in position $(x)$, but its detailed structure is different for different models. For RTP, it has the form (last line of Eq. (60) in the revised manuscript)
\begin{equation}
P_{\rm BL} = - \frac{v}{\omega}\,\frac{2\pi}{(4\pi D_e t)^{3/2}} \frac{v}{\sqrt{2\omega D_e}}\,x_0\,e^{-k_a x}\,e^{-\frac{x_0^2}{4D_e t}}\,. \nonumber
\end{equation}
We have now defined these in the text.

---

## Editorial Decision

resubmitted